# Integrated multi-omics analysis reveals the functional signature of microbes and metabolomics in pre-diabetes individuals

Yanmin Liu,[1] Qinwei Qiu,[1] Yang Chen,[1] Yusheng Deng,[1] Wei Huang,[1] Chen Sun,[1] Xiaoxiao Shang,[1] Xinyan Chen,[1] Chengrui Wang,[1] Lijuan Han,[2] Shiyan Chen,[3] Jiamin Yuan,[1] Fuping Xu,[1] Zhimin Yang,[1] Xiaodong Fang,[1] Li Huang[1]

**ABSTRACT** Pre-diabetes (PD) represents a critical stage in the progression toward type 2 diabetes, with significant alterations observed in the human microbial community among pre-diabetic individuals in observational studies. However, understanding the interaction between human microbiota and the host during pre-diabetes remains limited. Therefore, this study aims to understand the alterations in the human microbial community during pre-diabetes, a critical stage toward type 2 diabetes. Using an integrated analysis of human microbiota and metabolomics data, we seek to identify the functional signature associated with PD and gain insights into potential mechanisms driving its progression to type 2 diabetes. These findings could inform the development of early intervention strategies for those at high risk. Samples were collected from pre-diabetes, diabetes, and healthy control groups. Through metagenome and 16S rRNA sequencing, we analyzed the gut microbial and tongue coating compositions, respectively. Untargeted metabolomics techniques were also applied for comprehensive plasma data. Using integrated multi-omics analysis, we aim to understand the metabolic potentials of the human microbiome, its molecular links with host targets, and their effects on pre-diabetes, thereby deepening our understanding of microbiome-host interactions in this context. The pre-diabetes group exhibited distinct clinical characteristics, particularly in blood glucose levels and a higher average level of γ-glutamyl transferase. We identified 509 intestinal bacterial species, with *Megamonas funiformis* and *Parabacteroides merdae* showing higher abundance in the PD group. In tongue coating samples, we found 1,122 bacterial genera, with the PD group showing altered levels of *Corynebacterium* and *Johnsonella*. Furthermore, we detected 795 metabolites, primarily involved in carbohydrate and lipid metabolism. Importantly, our integrated multi-omics analysis suggested *Flavonifractor plautii*'s role in modulating blood glucose through influencing carbohydrate metabolism. Our integrated multi-omics analysis revealed significant alterations in several regulatory pathways associated with pre-diabetes, particularly emphasizing the impact of gut bacterium *Flavonifractor plautii* on blood glucose levels through its influence on carbohydrate metabolism. These intricate relationships among gut microbiota, metabolites, and blood glucose levels underscore the significance of personalized treatment approaches and preventive strategies for pre-diabetes. The insights gained from this research hold considerable promise for advancing our understanding and management of pre-diabetes.

**IMPORTANCE** This study investigates alterations in the human microbial community during PD, a critical stage leading to type 2 diabetes. Through integrated analysis of metagenomic and metabolomics data from pre-diabetes, diabetes, and healthy control groups, we identified distinct clinical characteristics in the PD group, including elevated blood glucose levels and γ-glutamyl transferase. A total of 509 intestinal bacterial species were identified, with *Flavonifractor plautii* playing a key role in modulating blood glucose levels via its influence on carbohydrate metabolism. Our findings underscore

**Peer Reviewer** Leo Gerlin, Eidgenossische Technische Hochschule Zurich, Zurich, Switzerland

Address correspondence to Xiaodong Fang, fangxd@gzucm.edu.cn, or Li Huang, liea1981@126.com.

Yanmin Liu and Qinwei Qiu contributed equally to this article. Author order was determined both alphabetically and in order of increasing seniority.

The authors declare no conflict of interest.

See the funding table on p. 16.

the complex interactions among gut microbiota, metabolites, and blood glucose levels, highlighting the potential for personalized treatment approaches and early intervention strategies for individuals at high risk of developing type 2 diabetes.

**KEYWORDS** pre-diabetes, multi-omics, *Flavonifractor plautii*, gut microbiota, oral microbiota, plasma metabolome

Pre-diabetes (PD) is a state characterized by elevated fasting blood glucose (GLU) levels that are higher than normal state but not reaching the threshold for diabetes diagnosis (1). It represents a high-risk condition for the development of type 2 diabetes. The global prevalence of pre-diabetes is currently increasing, and it is estimated that by 2045, the number of individuals affected by PD will exceed millions (2, 3). According to the Centers for Disease Control and Prevention, approximately 40% of adults in the United States have pre-diabetes, representing 88 million individuals (4). Similarly, a study conducted by China Medical University in 2020 revealed that the weighted prevalence of pre-diabetes in China was alarmingly high, reaching 35.2% (5). It is worth noting that approximately 5%–10% of individuals with PD will progress to type 2 diabetes each year, and ultimately, around 70% of individuals with PD will develop T2D (6–8).

In recent years, numerous studies have highlighted the significant association between human microbiota and pre-diabetes (9–13). For example, depletion of *Clostridium* and *Akkermansia muciniphila* has been observed in Danish adults with pre-diabetes (11). Additionally, a negative correlation was found between *Clostridium* spp. and fasting glucose levels in European women with normal, impaired, or diabetic glucose control (13). Another study revealed a decrease in *Clostridiales bacterium*, *Flavonifractor plautii*, and *Coprococcus eutactus* in the Swedish population with impaired fasting glucose (10). These findings suggest that the human microbiota plays a crucial role in maintaining host metabolic homeostasis and regulating glycemic levels, with variations in bacterial species observed across different regions. Zhong et al. have demonstrated that intestinal mucin proteins, such as MUC-1 and MUC-2, are present in fecal samples, which correlates with the relative abundance of *Akkermansia muciniphila* (12). In fact, *A. muciniphila* is a bacterium that primarily utilizes intestinal mucus as its carbon source. MUC1 and MUC2 are the main mucins secreted by intestinal epithelial cells, contributing to the formation of the intestinal mucus layer. Notably, *A. muciniphila* can effectively degrade these mucins to extract nutrients, a characteristic that plays a crucial role in supporting its growth and reproduction. Additionally, they also found that human proteins involved in glucose metabolism, including dipeptidyl peptidase 4 (DPP4), known to inhibit insulin secretion via its action on glucagon-like peptide-1 (GLP-1), tended to be lower in individuals with PD than in TN-T2D individuals (12). It is noteworthy that DPP4 produced by *Bacteroides* spp. has the ability to reduce the activity of GLP-1 and impair glucose metabolism in mice with intestinal permeability, which impacts the efficacy of sitagliptin (14), the current antidiabetic drug. These findings expand our understanding of the intestinal microbiota's role in regulating blood glucose levels. Ongoing advancements in research allow for the discovery of additional functions and mechanisms. However, our understanding of the functional characteristics and interactions between the human microbiota and the host remains limited. Further investigation is necessary to uncover more insights into the impact of the intestinal microbiota on blood glucose regulation.

Elucidating the mechanisms of host-microbial interaction in PD pathogenesis by integrating multi-omics data is highly significant. In this study, we identified and analyzed the characteristics of gut microbiome data, tongue coating 16S rRNA sequencing data, plasma metabolomics data, and clinical indicators from three distinct groups. Through further analysis, we revealed the ability of the gut bacterium *Flavonifractor plautii* to impact host blood glucose levels by influencing glucose metabolic pathways, thus providing new theoretical insights for the prevention and treatment of PD.

## MATERIALS AND METHODS

### Study design

To investigate the relationship between human microbiota and healthy status, we included 313 individuals (GDZYY-cohort 2021) aged 18-60 years in 2021. In addition to gathering clinical information through physical examination and fasting blood tests, we collected fecal samples for gut microbiome analysis, tongue coating samples for oral microbiome analysis, and serum samples for metabolome analysis. Here, we designed a PD case-control cohort based on their fasting blood glucose (Fbg) levels and divided into subgroups based on the 1999 World Health Organization (Geneva) criteria (15). Participants were divided into three groups based on their Fbg levels: diabetes group, defined as Fbg ≥7.0 mmol/L; pre-diabetes group, defined as Fbg ≥6.1 and <6.9 mmol/L; and the control group, defined as Fbg <6.1 mmol/L.

After performing exact matching for age and sex using the "matchit" function in the R package MatchIt (16), we enrolled a total of 31 healthy controls, seven pre-diabetic patients, and 10 diabetic patients in the study cohort. The flow diagram of enrolled individuals, including the number of each group, is shown in Fig. S1.

### Sample collection

Fresh stool samples were collected from the participants and were immediately frozen in a freezer at about −20°C. Frozen samples were transported to the laboratory using dry ice for 4 h. After delivery, the samples were stored at −80°C until DNA extraction. Total fecal genomic DNA was extracted from 0.5 g of fecal material.

Tongue coating samples were collected from the participants. Before the collection process, the oral cavity of each individual was washed with a normal saline solution two or three times. Sterile cotton swabs were utilized to collect samples from a specific region measuring 2 × 2 cm located in the central section of the dorsal surface of the tongue. Following 30 s of wiping, the samples were subsequently transferred into sterile test tubes and stored in a refrigerator set at −80°C until DNA extraction.

### Data source/measurement

DNA was quantified using the Qubit HS dsDNA kit (Invitrogen, Massachusetts, USA). A fixed aliquot of 50 ng DNA per sample was used to amplify the V4 region of the 16S rRNA gene with barcoded primers (515F/806R), following previous protocols. Pooled libraries, consisting of a mock community, two negative controls, and the samples, were loaded onto the Illumina MiSeq platform for obtaining 16S rRNA sequencing reads.

A metagenomic library, with an insert size of 350 bp, was constructed using high-quality DNA extracted from each sample. This was achieved using the TruSeq DNA PCR-Free Library Preparation Kit (Illumina, San Diego, USA). Subsequently, the library was sequenced on an Illumina NovaSeq platform (Illumina).

Metabolite concentrations were measured using an untargeted liquid chromatography-tandem mass spectrometry platform. To improve metabolite coverage, data from both positive and negative ions were collected using a high-resolution Q-Exactive HF mass spectrometer (Thermo Fisher Scientific, USA). The mass spectrometry raw data were preprocessed using Compound Discover (version 3.1) software (Thermo Fisher Scientific), then imported into metaX for normalization, batch effect correction, and compound filtration. Metabolite annotation was based on a combination of the BGI self-built standard library, mzCloud, and databases such as the Human Metabolome Database(HMDB) and Kyoto Encyclopedia of Genes and Genomes(KEGG).

### Data statistics

Statistical significance for the clinical data were assessed using two-sided Kruskal-Wallis tests. For the multivariate comparisons in microbiome analysis, permutational multivariate analysis of variance tests were performed, and the metabolome was analyzed using partial least squares discriminant analysis (PLS-DA).

The 16S rRNA amplicon sequencing reads were analyzed using Quantitative Insights into Microbial Ecology (QIIME2, version 2019.7) (17). Within QIIME2, the DADA2 software was employed to filter the sequencing reads and generate a table of microbial compositions. The Vsearch plugin was used to cluster sequences into operational taxonomic units (OTUs) at a 97% identity threshold, with taxonomy assignments made based on the Greengenes database (version 13.8).

The metagenomic shotgun sequencing data underwent preprocessing steps, including removal of adaptors, low-quality reads, and PCR duplicates from the raw reads. The remaining reads were then filtered to exclude host DNA using the hg19 (GRCh37) human reference genome. Metaphlan was utilized to establish the microbial composition, while HUMAnM2 was employed for Kyoto Encyclopedia of Genes and Genomes Ortholog (KO) assignment.

Statistical analyses were conducted using the R platform (version 3.6.1). In this study, PLS-DA and variable importance in projection (VIP) analysis were employed to analyze metabolomic data and identify significant metabolites contributing to the differences between groups. Prior to PLS-DA and VIP analysis, the metabolomic data underwent comprehensive pre-treatment to ensure data quality. This included data cleaning to remove noise and fill in missing values, which may have been caused by instrumental or sample processing errors. The data were then subjected to logarithmic transformation and standardization to eliminate dimensional differences between metabolites and to ensure the data distribution met the assumptions required for subsequent statistical analyses. Quality control was implemented using internal standards and blank controls to assess data accuracy and reproducibility, ensuring the reliability of the experimental data. Differential metabolites were defined based on a variable importance in projection value of $\geq 2$ from partial least squares discriminant analysis, as well as an adjusted $P$ value ($q$ value) of $\leq 0.05$ from the Wilcoxon test.

Differential taxonomic bacteria within the gut microbiota and tongue microbiota were determined using linear discriminant analysis effect size, which was implemented on the Galaxy platform (http://huttenhower.sph.harvard.edu/galaxy). Taxa showing linear discriminant analysis(LDA) scores of >2 and $P$ values of <0.05 were considered significant.

We implemented mediation analyses using mediate from the "mediation" package (18) to investigate the roles of human microbiota in PD through affecting blood glucose. The model-based causal mediation test was measured in two steps. In step 1, a mediator model and an outcome model were fitted. The mediator model was a linear regression of serum metabolite with age, sex, and human microbiota as the predictors (Model 1: metabolite = $\alpha 1 + \beta 1$ human microbiota i + $\delta 1 TXi + \varepsilon i1$). The outcome model was a linear regression model for Fbg phenotypes with the following covariates: human microbiota, human microbiota × serum metabolite interaction term, age, sex, and serum metabolite (Model 2: Fbg$i = \alpha 2 + \beta 2 \cdot$ human microbiota$i + \gamma 1 \cdot$ metabolite$i + \delta 2 \cdot$ TX$i + \varepsilon i2$). After the two models were fitted, the average causal mediation effect and average direct effect were computed through a general algorithm [$P = \beta 1\gamma 1 / (\beta 1\gamma 1 + \beta 2)$]. We used 1,000 iterations, and $P \leq 0.05$ was viewed as nominally significant.

## RESULTS

### Clinical signatures across pre-diabetes, control, and diabetes groups

In this study, we compared the clinical characteristics among three groups: the control group (control), the pre-diabetes group (pre-diabetes), and the diabetes group (diabetes). After matching for age and sex, we investigated differences in various clinical biochemical markers, as presented in Table 1.

In the pre-diabetes group, significant differences were observed in several clinical characteristics compared to the control group. Blood glucose concentration increased sequentially from the control group to the diabetes group ($P < 0.001$) (Fig. 1A), indicating impaired glucose metabolism. Additionally, serum γ-glutamyl transferase (GGT) concentrations were significantly increased in both the pre-diabetes and diabetes groups

**TABLE 1** The comparison of clinical characteristics among the three groups[a]

|  | Control | Pre-diabetes | Diabetes | P |
| --- | --- | --- | --- | --- |
| Age (mean [SD]) | 39.87 (6.35) | 39.57 (5.35) | 44.8 (7.74) | 0.111 |
| Sex: male (%) | 13.0 (41.9) | 3.0 (42.9) | 5.0 (50) | 0.904 |
| Waistline (mean [SD], cm) | 73.27 (18.66) | 60.71 (26.92) | 75.4 (29.96) | 0.441 |
| Hipline (mean [SD], cm) | 90.61 (16.24) | 74.5 (29.51) | 97.0 (11.52) | 0.169 |
| SBP (mean [SD]) | 116.42 (12.24) | 116.33 (10.39) | 120.4 (10.16) | 0.783 |
| DBP (mean [SD]) | 72.21 (9.57) | 69.33 (5.16) | 77.8 (4.21) | 0.247 |
| UA (mean [SD], μmol/L) | 325.19 (116.49) | 351.86 (106.48) | 353.9 (94.54) | 0.709 |
| GLU (mean [SD], mmol/L) | 5.2 (0.52) | 6.41 (0.21) | 8.4 (1.16) | <0.001 |
| LDLC (mean [SD], mmol/L) | 2.94 (0.73) | 2.97 (0.64) | 3.5 (0.98) | 0.151 |
| TC (mean [SD], mmol/L) | 4.87 (0.64) | 4.6 (0.8) | 5.25 (1.2) | 0.244 |
| TG (mean [SD], mmol/L) | 1.34 (1.39) | 1.42 (0.95) | 1.78 (0.83) | 0.621 |
| Non-HDLC (mean [SD], mmol/L) | 3.44 (0.75) | 3.44 (0.81) | 4.11 (1.15) | 0.098 |
| HDLC (mean [SD], mmol/L) | 1.43 (0.35) | 1.16 (0.26) | 1.14 (0.24) | 0.019 |
| CRP (mean [SD]) | 0.79 (1.44) | 1.26 (1.34) | 0.97 (1.48) | 0.741 |
| NEU % (mean [SD]) | 57.22 (6.75) | 55.83 (9.45) | 59.91 (5.4) | 0.443 |
| NEU (mean [SD]) | 3.25 (0.91) | 3.31 (1.42) | 4.17 (1.02) | 0.053 |
| MONO % (mean [SD]) | 6.83 (1.69) | 6.91 (1.22) | 6.61 (1.69) | 0.913 |
| MONO (mean [SD]) | 0.39 (0.14) | 0.41 (0.18) | 0.46 (0.14) | 0.443 |
| BASO% (mean [SD]) | 0.56 (0.3) | 0.4 (0.19) | 0.58 (0.36) | 0.403 |
| BASO (mean [SD]) | 0.03 (0.02) | 0.03 (0.02) | 0.04 (0.02) | 0.42 |
| EOSIN % (mean [SD]) | 2.39 (1.62) | 2.79 (1.4) | 1.78 (1.13) | 0.369 |
| EOSIN (mean [SD]) | 0.14 (0.11) | 0.18 (0.12) | 0.13 (0.09) | 0.605 |
| MCHC (mean [SD]) | 326.29 (16.35) | 331.14 (9.46) | 330.5 (13.54) | 0.617 |
| MCH (mean [SD]) | 28.63 (3.82) | 30.2 (1.95) | 29.21 (4.13) | 0.586 |
| MCV (mean [SD]) | 87.44 (8.51) | 91.21 (4.87) | 88.06 (9.35) | 0.558 |
| MPV (mean [SD]) | 10.17 (0.84) | 9.87 (0.56) | 9.94 (1.08) | 0.607 |
| LYM. (mean [SD]) | 32.99 (6.68) | 34.07 (9.84) | 31.12 (4.45) | 0.649 |
| LYM (mean [SD]) | 1.84 (0.47) | 2.02 (0.95) | 2.16 (0.58) | 0.297 |
| WBC (mean [SD]) | 5.65 (1.27) | 5.95 (2.4) | 6.95 (1.56) | 0.076 |
| RDW (mean [SD]) | 12.72 (1.54) | 12.03 (0.39) | 12.67 (1.83) | 0.544 |
| HCT (mean [SD]) | 42.22 (5.07) | 43.2 (5.24) | 42.85 (3.44) | 0.858 |
| RBC (mean [SD]) | 4.86 (0.64) | 4.77 (0.76) | 4.94 (0.87) | 0.885 |
| PDW (mean [SD]) | 11.44 (1.9) | 10.93 (1.38) | 11.41 (2.88) | 0.838 |
| PLT (mean [SD]) | 244.61 (54.77) | 269.29 (73.5) | 274.3 (82.16) | 0.361 |
| Hb (mean [SD],) | 138.13 (20.33) | 143.14 (18.89) | 141.4 (10.82) | 0.765 |
| TSH (mean [SD], mIU/L) | 1.93 (0.96) | 1.72 (0.53) | 2.09 (1.02) | 0.714 |
| FT3 (mean [SD], pmol/L) | 5.11 (0.57) | 5.8 (0.8) | 5.28 (0.47) | 0.024 |
| FT4 (mean [SD], pmol/L) | 15.2 (2.22) | 16.5 (1.65) | 16.07 (2.2) | 0.261 |
| AFP (mean [SD], ng/mL) | 2.57 (1.63) | 2.44 (0.55) | 2.4 (0.56) | 0.935 |
| CEA (mean [SD], ng/mL) | 1.76 (1.2) | 1.41 (0.61) | 2.02 (1.03) | 0.535 |
| ALB_GLB (mean [SD]) | 1.93 (0.34) | 1.83 (0.18) | 1.99 (0.47) | 0.652 |
| GGT (mean [SD], U/L) | 19.97 (10.74) | 26.0 (8.96) | 48.5 (38.83) | 0.001 |
| TBA (mean [SD], μmol/L) | 3.94 (3.04) | 4.4 (2.54) | 4.37 (2.54) | 0.88 |
| TP (mean [SD], g/L) | 73.59 (3.73) | 76.23 (3.16) | 74.8 (5.16) | 0.263 |
| GLB (mean [SD], g/L) | 25.58 (3.84) | 27.14 (2.63) | 25.81 (5.39) | 0.659 |
| ALB (mean [SD], g/L) | 48.01 (2.65) | 49.09 (1.96) | 48.99 (2.56) | 0.421 |
| ALT (mean [SD], U/L) | 18.23 (11.84) | 21.0 (10.54) | 26.0 (19.37) | 0.294 |
| AST (mean [SD], U/L) | 18.13 (5.12) | 20.57 (5.65) | 22.1 (12.23) | 0.288 |
| AST_ALT (mean [SD]) | 1.23 (0.53) | 1.16 (0.47) | 0.98 (0.34) | 0.393 |
| Urea (mean [SD], mmol/L) | 4.71 (1.16) | 4.57 (0.76) | 4.97 (0.96) | 0.725 |
| Cr (mean [SD], μmol/L) | 71.39 (17.17) | 71 (12.03) | 75.4 (14.93) | 0.776 |

TABLE 1 The comparison of clinical characteristics among the three groups[a] (*Continued*)

|  | Control | Pre-diabetes | Diabetes | *P* |
|---|---|---|---|---|
| eGFR (mean [SD], mL/min/1.73 m$^2$) | 102.93 (12.57) | 104.51 (7.3) | 96.58 (13.41) | 0.303 |

[a]AFP, alpha-fetoprotein; ALB, albumin; ALT, alanine aminotransferase; AST, aspartate aminotransferase; BASO, basophil; CEA, carcinoembryonic antigen; Cr, creatinine; CRP, C-reactive protein; DBP, diastolic blood pressure; eGFR, estimated glomerular filtration rate; EOSIN, eosinophil; FT3, free triiodothyronine 3; FT4, free triiodothyronine 4; GGT, gamma-glutamyltransferase; GLB, globulin; GLU, glucose; Hb, hemoglobin; HCT, hematocrit; LDLC, low-density lipoprotein cholesterol; LYM, lymphocyte; MCH, mean corpuscular hemoglobin; MCHC, mean corpuscular hemoglobin concentration; MCV, mean corpuscular volume; MONO, monocyte; MPV, mean platelet volume; NEU, neutrophil; PDW, platelet distribution width; PLT, platelet; RBC, red blood cell; RDW, red cell distribution width; SBP, systolic blood pressure; TBA, total bile acid; TC, total cholesterol; TG, total cholesterol; TP, total protein; TSH, thyroid-stimulating hormone; UA, uric acid; WBC, white blood cell.

($P < 0.05$) (Fig. 1B), which is indicative of liver dysfunction (19, 20). Interestingly, our results demonstrate that GGT concentrations are significantly elevated in the PD stage compared to the normal stage, with no further significant increase observed between the PD and diabetes stages. In contrast, blood glucose concentration shows a continuous rise from the normal stage through the PD stage and into the diabetes stage. This pattern implies that hepatic dysfunction, as indicated by the elevated GGT concentrations, is established during the PD stage and may precede the subsequent increase in blood glucose levels. These findings highlight the potential for early liver involvement in metabolic dysregulation and support the notion that alterations in hepatic function play a critical role in the pathogenesis of glucose metabolism disorders. Furthermore, they underscore the potential utility of GGT as a valuable biomarker for identifying individuals at risk of developing diabetes. Furthermore, compared to the control group, the diabetes group showed a significant reduction in HDLC levels (Fig. 1D, $P < 0.05$). The Pre-diabetes group had HDLC levels comparable to the Diabetes group and significantly lower than the Control group, though this difference lacked statistical significance (Fig. 1D). This finding is strongly associated with impaired fasting blood sugar and diabetes, indicating dyslipidemia among individuals with pre-diabetes (21). Notably, free triiodothyronine (FT3) levels increased only in the pre-diabetes group, with no significant difference observed between the control and diabetes groups (Fig. 1C, $P < 0.05$).

In summary, significant differences in specific clinical biochemical markers were observed among individuals with pre-diabetes compared to the control group and diabetes group. These findings provide valuable insights into the biochemical alterations occurring during the progression from normal glucose tolerance to pre-diabetes and diabetes.

## Alterations in gut microbiota composition associated with pre-diabetes and diabetes

Fecal samples were collected from participants, and metagenomic methods were employed to analyze the microbial composition. The intestinal microbial profiles of the control, pre-diabetes, and diabetes groups are presented in Fig. 2A, revealing the presence of *Bacteroides* and *Prevotella* in all three groups. Notably, various species of *Bacteroides*, including *Bacteroides vulgatus*, *Bacteroides uniformis*, and predominantly *Prevotella copri*, were identified, which is consistent with findings from previous studies on intestinal microbiota composition (22, 23). These preliminary results support the reliability of our data. Further analysis of the relative abundance of species among different groups demonstrated that *Megamonas funiformis* was significantly enriched primarily in the pre-diabetes group compared to both the control and diabetes groups. *M. funiformis* is an anaerobic bacterium belonging to the family *Veillonellaceae*, thriving in low-oxygen environments (24, 25). Its enrichment in individuals with pre-diabetes or specific metabolic conditions suggests its potential as a biomarker for pre-diabetes (26, 27). In fact, *M. funiformis* has been identified in various studies examining its potential impact on metabolic pathways. Its presence may correlate with improved metabolic profiles, including better glucose regulation. High-fiber diets promote the growth of beneficial gut bacteria, which in turn can lead to improved glycemic control (28). This

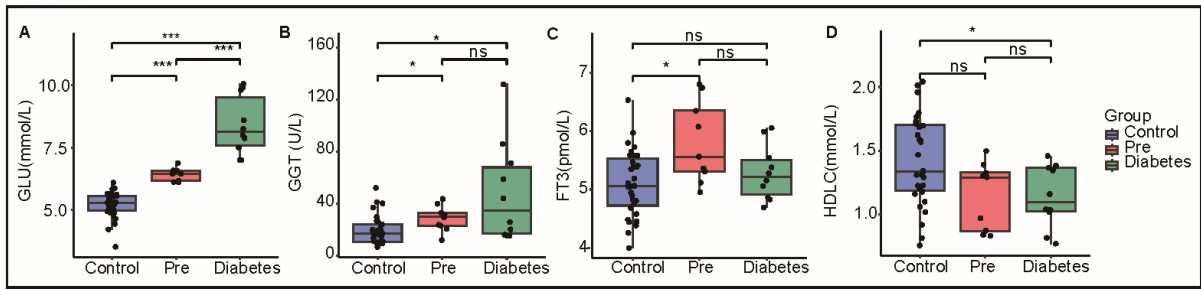

**FIG 1** Comparison of the clinical signatures between the pre-diabetes, control, and diabetes groups (A–D). Boxplot shows the clinical characteristic index between different groups, including blood glucose indicators (A), GGT (B), FT3 (C), and HDLC (D). The points indicate each sample, and the blue box represents the control group; the red box represents the pre-diabetes group; and the green box represents the diabetes group. ***$P < 0.001$, *$P < 0.05$ $P$ values refer to Student's two-sided $t$-tests for statistical significance. FT3, free triiodothyronine; GGT, γ-glutamyl transferase; GLU, glucose; HDLC, high-density lipoprotein cholesterol; ns, non-significant.

relationship underscores the importance of understanding how specific gut microbes, such as *Megamonas funiformis*, contribute to metabolic health and the potential for targeted dietary strategies to manage blood glucose levels.

In the control group, significant enrichment was observed for *Clostridium bolteae* (Fig. 2B) and *Flavonifractor plautii* (Fig. 2D). The abundance of *C. bolteae* decreased in the pre-diabetes group but remained relatively stable in both the diabetes and control groups. Conversely, *F. plautii* abundances decreased with increasing blood glucose levels (Fig. 2J), indicating a close association that *F. plautii* may play a role in regulating blood glucose due to its potential to produce butyrate (29, 30). Additionally, we noted an increase in the abundance of *Bacteroides xylanisolvens* in the diabetes groups. This species has been reported to degrade nicotine, produce folic acid, and participate in host metabolism (31).

In summary, our study revealed significant differences in the microbial composition among the control, pre-diabetes, and diabetes groups. *M. funiformis* was notably enriched in the pre-diabetes group, while *C. bolteae* and *F. plautii* showed significant enrichment in the control group. The abundance of *F. plautii* was negatively correlated with blood glucose levels, suggesting its potential involvement in blood glucose regulation through butyric acid production. Furthermore, *Bacteroides xylanisolvens* exhibited an increase in abundance in the diabetes groups. These findings illuminate the potential influence of gut microbiota on pre-diabetes and provide avenues for further research.

## Oral microbial composition changes in pre-diabetes

Tongue coating samples were collected from the enrolled participants, and 16S rRNA sequencing was conducted to analyze the oral microbial composition. A total of 1,122 bacterial genera were detected, among which 165 genera were present in at least 10% of the samples. Notably, *Prevotella*, *Neisseria*, *Haemophilus*, *Alloprevotella*, *Fusobacterium*, and *Porphyromonas* were identified as common bacteria in tongue coating (Fig. 3A).

Of particular interest, the abundance of *Corynebacterium* was significantly lower in the pre-diabetes group compared to both the control and the diabetes groups (Fig. 3B). *Corynebacterium* is a genus of gram-positive bacteria, rod-shaped bacteria belonging to the phylum *Actinobacteria*, commonly found in the human oral cavity (32). These species are known for their diverse metabolic capabilities and are part of the commensal organisms in the human microbiota. The reduced level of oral *Corynebacterium* observed in the pre-diabetes group suggests a potential association between the decreased abundance of this genus and the development or progression of pre-diabetes. While the exact role of *Corynebacterium* in pre-diabetes remains unclear, further research is needed to elucidate its functional significance. It has been proven that *Corynebacterium* may

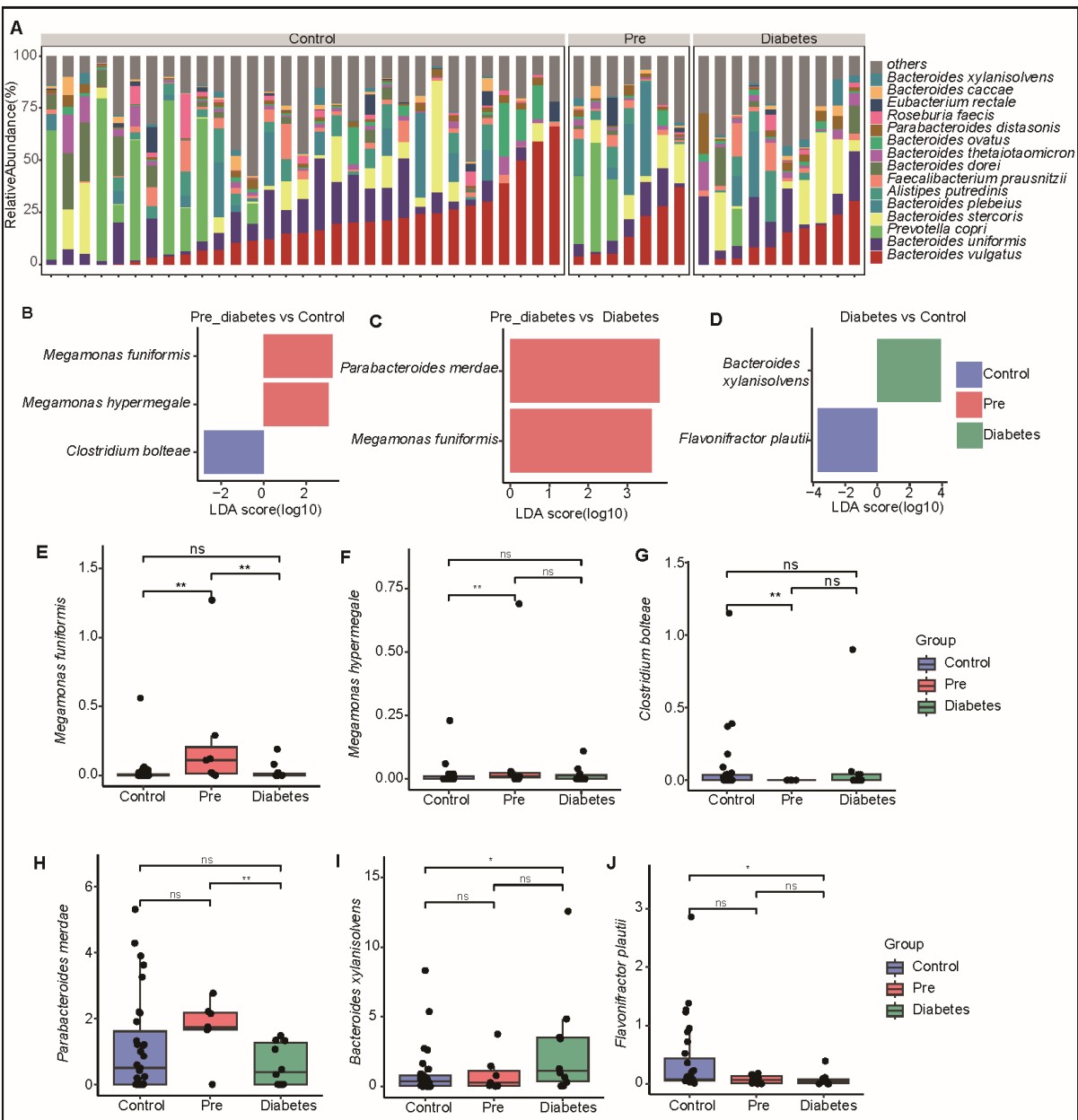

FIG 2 Insights into the gut microbial characteristics in pre-diabetes (A) Bar bacterial composition at the species level of fecal samples from different groups: control, pre-diabetes, and diabetes groups. The top 15 bacterial species are displayed, while the rest are categorized as "others." (B through D) LEfSe analysis of the gut microbial composition of the three study groups (LEfSe analysis exhibited differentially abundant taxa with $P < 0.05$ and linear discriminant analysis score >2.0). LEfSe analysis results between the pre-diabetes and control groups (B); LEfSe analysis results between the pre-diabetes vs diabetes groups (C); LEfSe analysis results between the diabetes and control groups (D). (E through J) Boxplot displays the relative abundance of bacteria in each group, including *Megamonas funiformis* (E), *Megamonas hypermegale* (F), *Clostridium bolteae* (G), *Parabacteroides merdae* (H), *Bacteroides xylanisolvens* (I), and *Flavonifractor plautii* (J). The points indicate each sample, and the blue box represents the control group; the red box represents the pre-diabetes group; and the green box represents the diabetes group. **$P < 0.01$, *$P < 0.05$. ns, non-significant.

play a role in maintaining oral health and contributing to the overall balance of the oral microbiota (33, 34). Additionally, exploring potential interactions between *Corynebacterium* and other microbial species in the oral ecosystem could further insights into their collective influence on oral health and systemic health outcomes related to pre-diabetes. Conversely, the increased abundance of *Johnsonella* in the pre-diabetes group compared

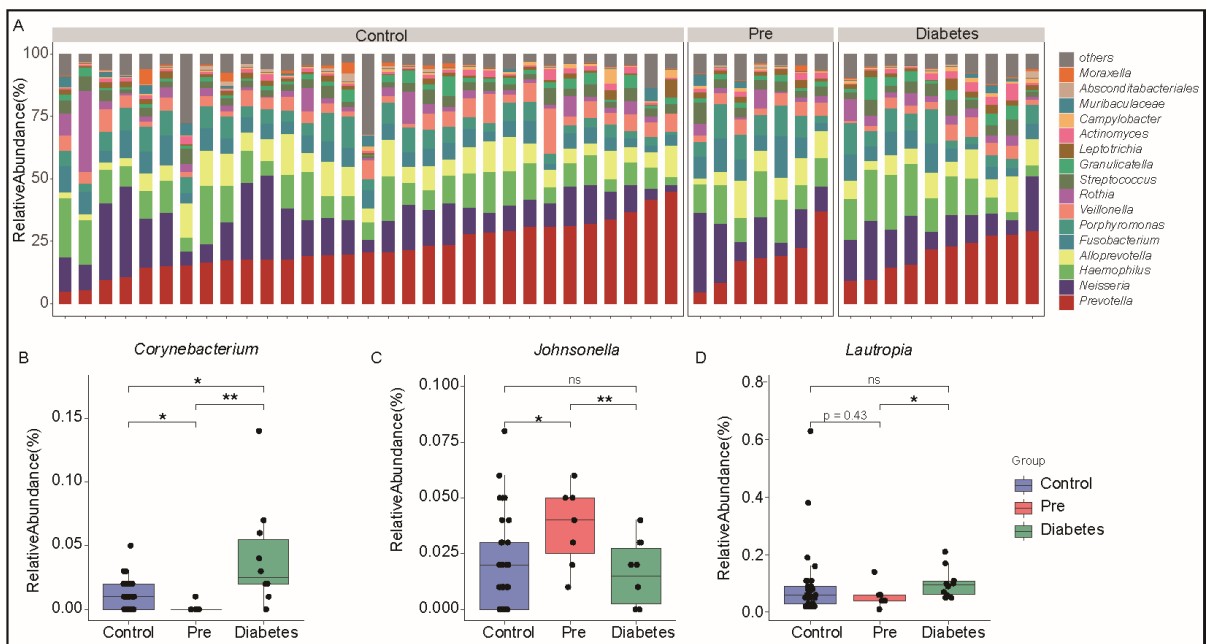

**FIG 3** Exploring tongue coating microbial characteristics in pre-diabetes (A) Bar bacterial composition at the genus level of tongue coating samples from different groups: control, pre-diabetes, and diabetes groups. The top 15 bacterial species are displayed, while the rest are categorized as "others." (B through D) Boxplot displays the relative abundance of bacteria in each group, including *Corynebacterium* (B), *Johnsonella* (C), and *Lautropia* (D). The points indicate each sample, and the blue box represents the control group; the red box represents the pre-diabetes group; and the green box represents the diabetes group. ***$P < 0.001$, **$P < 0.01$, *$P < 0.05$. ns, non-significant.

to both the control and the diabetes groups indicates its potential involvement in pre-diabetes (Fig. 3B). *Johnsonella*, a gram-negative, rod-shaped bacterium belonging to the phylum Bacteroidetes, has been identified in various ecological niches, including the human oral cavity, and is part of the commensal microbiota (35, 36).

These findings highlight specific alterations in the tongue coating microbiota associated with pre-diabetes, emphasizing the potential role of these bacterial genera in oral health and disease development. Overall, *Corynebacterium* and *Johnsonella* represent intriguing microbial genera within the oral microbiota, warranting further investigation to unravel their potential contributions to disease progression and management. Further exploration of the functional roles of *Corynebacterium* and *Johnsonella*, as well as their interactions within the oral microbiota, may provide valuable insights into the pathogenesis and progression of pre-diabetes.

## Metabolomic alterations in pre-diabetes: key insights into glucose metabolism

We compared the serum metabolome data among the control, pre-diabetes, and diabetes groups. PLS-DA revealed significant differences in metabolite composition between these groups (Fig. 4A). In addition, our metabolomic analyses revealed significant differences in metabolic profiles among the diabetes group, healthy control group, and pre-diabetes group (Fig. S1A and C). The key differentially abundant metabolites distinguishing the healthy controls from diabetes patients were ethyl oleate, followed sequentially by dodecylbenzene sulfonic acid, fluvoxamine, and PS20 (Fig. S1B). Notably, the critical metabolic distinctions between the diabetes group and the pre-diabetes group were characterized by differential abundance of pyrimidine, succinic acid semialdehyde, and biliverdin (Fig. S1D). The key metabolites that contribute most to the discrimination between the pre-diabetes group and control group include glyceraldehyde, L-gulose, N-methyl L-glutamic acid, benzene triol, hydroxymethylglutaric

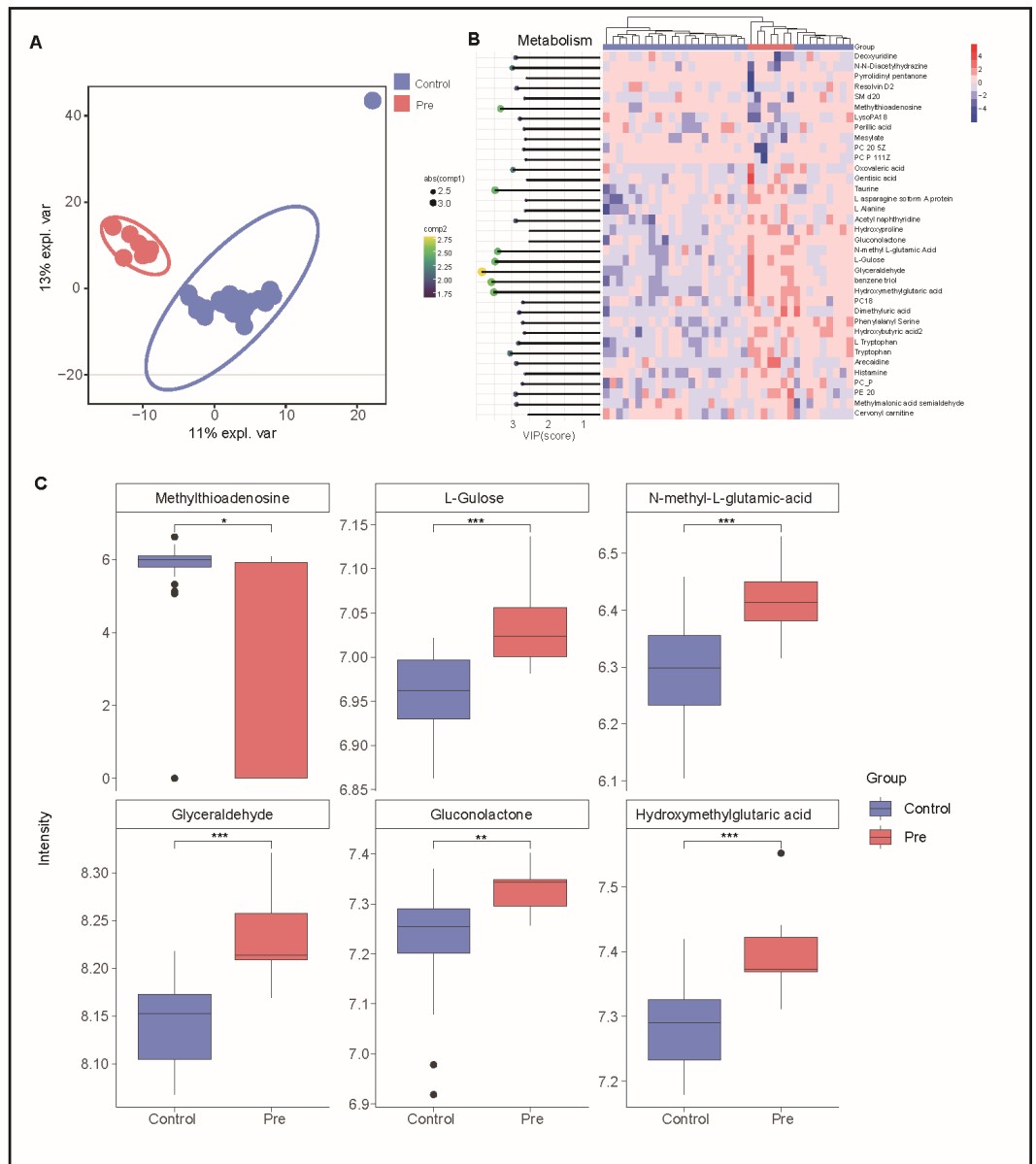

**FIG 4** Metabolic alterations in the glucose pathway distinguish pre-diabetes from control groups. (A) Partial least squares discriminant analysis (PLSDA) plot with samples' identification, showing the discrimination between the control and pre-diabetes groups (red denotes pre-diabetes; blue denotes control). (B) VIP scores derived from PLSDA loadings to show the importance of each metabolite. The heatmap displays the abundance of metabolites in the control and pre-diabetes groups, with the rows that are scaled and clustered using Euclidean distance measurement. The color of the space ranges from red to blue. Redder spaces indicate a greater abundance of metabolite in the sample. (C) Boxplot shows abundance of metabolites that play an important role in discrimination of pre-diabetes and control groups. *P* values refer to Wilcoxon tests for statistical significance.***$P < 0.001$, **$P < 0.01$, *$P < 0.05$.

acid, taurine, and methylthioadenosine (Fig. 4B). Of them, glyceraldehyde is the most important intermediate in glucose metabolism and glycolysis. Glyceraldehyde is derived from the glucose breakdown and plays a crucial role in energy production within cells. Our data indicated significant positive correlations between glyceraldehyde levels and blood glucose levels in individuals with pre-diabetes, suggesting that elevated glyceraldehyde concentrations may contribute to impaired glucose metabolism and the development of hyperglycemia. Understanding the role of glyceraldehyde in pre-diabetes is crucial for elucidating the mechanisms underlying glucose dysregulation and for

identifying potential intervention or therapeutic targets. Further studies are warranted to explore its precise implications and its potential as a therapeutic target in glucose homeostasis and pre-diabetes management.

Additionally, we identified elevated levels of several metabolites in the pre-diabetes group, which play important roles in glucose metabolism and directly influence this fundamental metabolic pathway. Specifically, increased levels of hydroxy-methyl-glutaric acid, L-gulose, and gluconolactone were observed (Fig. 4C). Hydroxy-methyl-glutaric acid participates in various cellular processes, including energy metabolism (37), and serves as an intermediate in the breakdown of amino acids and fatty acids. The increased levels of hydroxy-methyl-glutaric acid in individuals with pre-diabetes suggest perturbations in these metabolic pathways, potentially contributing to dysregulated glucose metabolism. L-Gulose, a rare six-carbon sugar that can be converted to L-galactose in one step (38), is involved in glycolysis, the pathway responsible for breaking down glucose to produce energy. Increased levels of L-gulose in individuals with pre-diabetes indicate alterations in glucose metabolism and highlight potential disturbances in glycolytic processes. Gluconolactone, derived from glucose, is involved in various metabolic reactions, including carbohydrate metabolism, and serves as an intermediate in converting glucose to gluconic acid. Elevated levels of gluconolactone may signal imbalances in glucose metabolism and disruptions in glucose as an energy source. Furthermore, our findings indicate an increase in N-methyl-L-glutamic acid in the pre-diabetes group (Fig. 4C). Prior studies have demonstrated that this metabolite exerts regulatory effects on glucose metabolism (39, 40), influencing insulin secretion, insulin resistance, and energy metabolism, thereby impacting overall glucose homeostasis. It is noteworthy that there is an outlier in the control group. To assess its impact on the overall analysis, we first analyzed the metabolic signature of the outlier, which is characterized by significantly reduced levels of specific metabolites, including N-acetyl-glutamine, L-tyrosine, and pyrrolidine. Subsequently, we reperformed statistical analyses excluding this sample. As shown in Fig. S2, the overall trends remained largely consistent. Glyceraldehyde continued to be the primary differentiator between the two groups. The order of contribution from other metabolites and the abundance of differential metabolites in Fig. 4C also stayed the same. Moreover, this result confirmed that including this outlier did not alter our fundamental conclusion.

In summary, metabolites enriched in the glucose metabolism pathway are significantly altered in the pre-diabetes group compared to the control group. These findings underscored the disrupted glucose metabolism present in the pre-diabetes group, contributing to the reversible elevation of blood glucose levels.

## Gut microbiota and metabolite impact on blood glucose in pre-diabetes

In our study, we aimed to investigate the mechanisms underlying remission in patients with pre-diabetes through the integration of multi-omics data. To achieve this, we employed various analytical approaches, including correlation analysis, network analysis, and mediation analysis, to explore the complex relationships within the multi-omics data set. Correlation analyses revealed significant positive associations between several metabolites and blood glucose levels. Notably, these metabolites included glyceraldehyde, hydroxy-methyl-glutaric acid, L-gulose, gluconolactone, gluconic acid, and galacturonic acid, all of which play direct or indirect roles in glucose metabolism (Fig. 4A).

Furthermore, we observed a positive correlation between the gut bacterium *Bacteroides xylanisolvens* and blood glucose levels, while *Flavonifractor plautii* exhibited an inverse correlation (Fig. 5A). These findings highlight the intricate interplay between various metabolites and blood glucose regulation in individuals with pre-diabetes. The identified positive correlations suggest that these metabolites may significantly impact glucose metabolism, emphasizing their relevance in pre-diabetic conditions. Importantly, these results contribute to a broader understanding of the interactions

within the metabolic network and highlight potential biomarkers or therapeutic targets for pre-diabetes management.

We also focused on serum GGT, which demonstrated strong correlations with several clinical biochemical markers (Fig. 5B), suggesting its potential role in elevating blood glucose levels. There are three primary reasons for selecting GGT as a marker: first, GGT levels show significant changes in clinical indicator analysis alongside GLU levels, with notably elevated levels observed in both the PD group and the diabetic group compared to the Control group (see Fig. 1B). Second, our correlation analysis revealed that GGT exhibited the strongest associations with other omics data (Fig. 5B). Finally, GGT plays an important role in glucose metabolism, further substantiating its relevance as a marker in our study.

To examine our hypothesis that gut microbes may influence blood glucose levels through the production or consumption of specific metabolites, thereby affecting the pathogenesis and reversal of pre-diabetes, we employed a stepped-mediation analysis model. This model integrated intestinal metagenomics, serum metabolomics, blood glucose levels, and other relevant data. Our analysis provided compelling evidence for the significant role of *F. plautii* in modulating blood glucose levels, with serum glyceraldehyde acting as a key mediator (Fig. 5C). Notably, we observed a negative correlation between the abundance of *F. plautii* and glyceraldehyde levels (Fig. 5D). Additionally, glyceraldehyde showed a positive correlation with GLU levels (Fig. 5E) and serum GGT levels (Fig. 5F). These findings illustrate the potential impact of *F. plautii* on blood glucose regulation, highlighting glyceraldehyde's role as an intermediary molecule. The negative correlation between *F. plautii* abundance and glyceraldehyde levels suggests that *F. plautii* may influence glucose metabolism by modulating glyceraldehyde production or utilization. Furthermore, the positive correlations between glyceraldehyde, GLU levels, and GGT further emphasize glyceraldehyde's relevance in the context of pre-diabetes. This underscores the intricate interplay between the gut microbiota and host metabolism, providing valuable insights into potential therapeutic targets for managing blood glucose dysregulation.

In conclusion, our population-based data elucidate the complex relationships among gut microbiota, specific metabolites, and blood glucose levels. Our study contributes to a deeper understanding of the mechanisms underlying pre-diabetes development and highlights potential intervention strategies. These findings pave the way for future research aimed at unraveling the precise molecular pathways involved and exploring the therapeutic potential of targeting the gut microbiota-metabolite-glucose axis.

## DISCUSSION

Our findings highlight significant alterations in the human microbial community during pre-diabetes, a crucial stage in the progression toward type 2 diabetes. By integrating analysis of human microbiota and metabolomic data, we identified functional signatures and potential mechanisms associated with pre-diabetes. This comprehensive approach enhanced our understanding of the metabolic capacities of the human microbiome, its molecular connections with host targets, and their effects on pre-diabetes. Future research should focus on validating these findings in larger cohorts and exploring the therapeutic implications of modulating gut microbiota for blood glucose regulation. We found that FT3 levels are increased only in the pre-diabetes group. Both FT3 and FT4 are active forms of thyroid hormones, with FT3 exerting stronger physiological effects. Thyroid hormones play an essential role in regulating glucose metabolism, and increased blood glucose levels during fasting and postprandial states have been noted in patients with hyperthyroidism due to the exacerbated insulin resistance caused by thyroid hormones (41–43).

Intriguingly, our multi-omics analysis revealed the important role of the gut bacterium *Flavonifractor plautii* in regulating blood glucose levels through its impact on carbohydrate metabolism. This suggests a complex interplay between gut microbiota, metabolic pathways, and host physiological responses. Uncovering these intricate

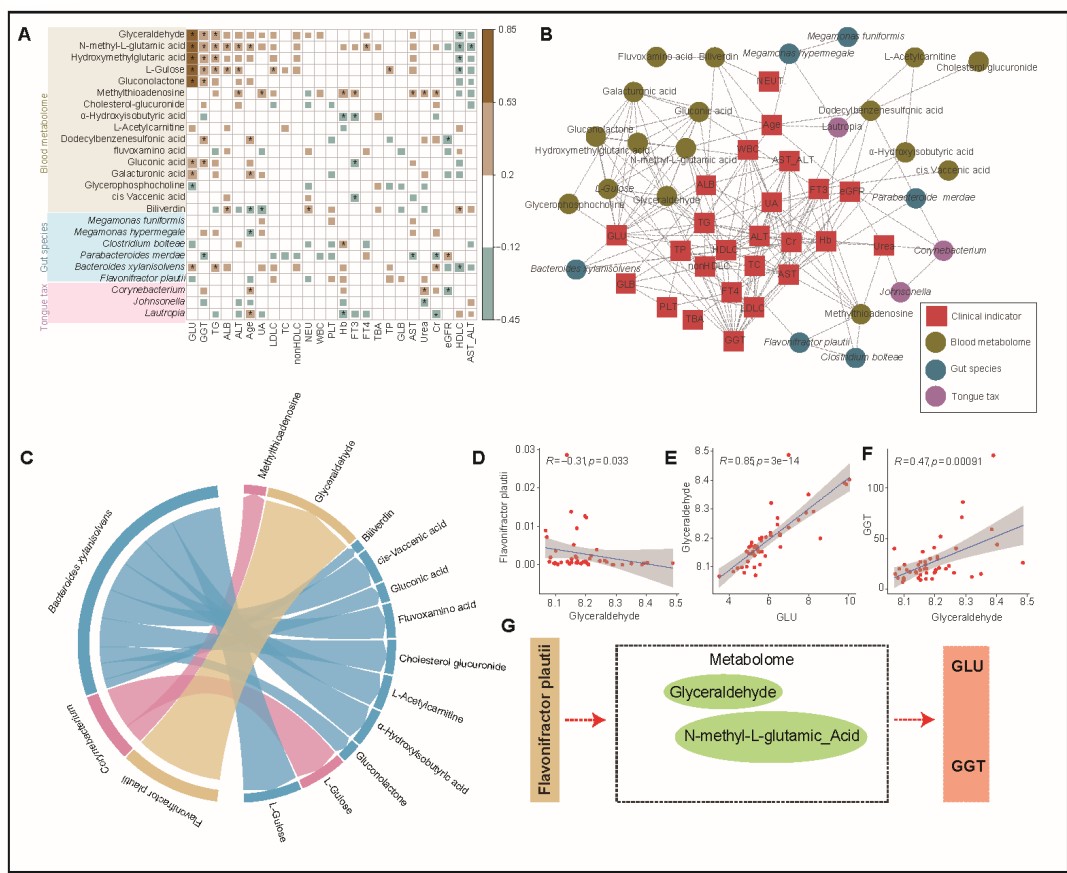

**FIG 5** Integrating multi-omics data reveals mechanisms of remission in pre-diabetes patients (A) Heat map showing the associations of clinical characteristics with the intestinal microbiota, tongue coating microbiota, and plasma metabolite, calculated by using Spearman's correlation. These features of microbiome and metabolites are identified as being enriched or decreased in pre-diabetic patients. Brown values indicate features were positively correlated with clinical data, while green ones indicate the features were negatively correlated with clinical data. Significant associations (adjusted $P < 0.05$) are indicated by asterisks. (B) The association between features and clinical characteristics. The circle represents the features, including intestinal microbiome (blue), tongue coating microbiome (purple), and plasma metabolites (brown); the square represents the clinical characteristics. (C) Circos plot illustrating the mediation effect of the microbiome feature between blood glucose levels. Each band linking a microbiome feature and a metabolite represents a mediation of the microbiome feature between the metabolites and its associated blood glucose levels. The width of the band is proportional to the $P$ value in the forward mediation. Microbiome features are colored according to their $z$-scores in association with the corresponding exposure factors. (D) Spearman's test showed that the relative abundance (log) of *Flavonifractor plautii* was negatively correlated with glyceraldehyde level. (E) Spearman's test showed that the glyceraldehyde level was positively correlated with blood glucose levels. (F) Spearman's test showed that the glyceraldehyde level was positively correlated with gamma-glutamyltransferase (GGT). (G) Highlighted examples of microbiome *Flavonifractor plautii* mediators between plasma metabolites and blood glucose level.

relationships underscores the significance of personalized treatment approaches and preventive strategies for pre-diabetes. Supporting this, a recent study demonstrated the role of gut microbiota in blood glucose regulation. For example, certain gut bacteria, including those from the Bacteroidetes phylum, have been shown to produce short-chain fatty acids that can enhance insulin sensitivity and modulate glucose metabolism (44). Moreover, the interplay between dietary fibers and gut microbiota, including *Megamonas funiformis*, suggests that dietary interventions could influence blood glucose levels. Butyric acid, a short-chain fatty acid generated by bacterial fermentation of carbohydrates, can activate G protein-coupled receptors GPCR41 and GPCR43 in the gut, stimulating the release of hormones such as GLP-1 in intestinal epithelial L cell (45–48). GLP-1 primarily regulates blood glucose levels by promoting insulin production, enhancing insulin sensitivity, and alleviating insulin resistance (49–51). Therefore, increasing the presence of butyrate-producing bacteria may improve

insulin sensitivity and metabolic abnormalities, providing a theoretical basis for further investigation into the impact of *F. plautii* and butyric acid production on the pathogenesis of pre-diabetes.

Through multi-omics analysis, strains such as Alistipes, *Bacteroides*, and *Flavonifractor* strains were found to be negatively correlated with carbohydrate levels in the body, indicating their contribution to reducing carbohydrate levels in the host. *In vitro* experiments showed that only *Alistipes indistinctus* has direct carbohydrate consumption, including glucose, fructose, and mannose, while *Bacteroides* and *Flavonifractor* have limited abilities in directly consuming carbohydrates (52). Another study found that *Bacteroides* could ameliorate obesity, insulin resistance, and high-fat diet-induced intestinal permeability in mice by producing N-acetylglucosamine and enriching *Akkermansia muciniphila* (53). Notably, *Akkermansia muciniphila* stimulates GLP-1 secretion in intestinal L cells through P9 protein production and binding to ICAM-2, promoting thermogenesis in brown adipose tissue and improving glucose homeostasis in mice (54). However, the specific mechanism by which *Flavonifractor plautii* affects blood glucose is not yet fully understood. Some literature indicates that *F. plautii* abundance is higher in the pre-diabetes group and negatively correlated with body fat, alongside demonstrating a weaker glycolytic capacity in the population with increased arterial stiffness (55). Functional analysis revealed a decrease in the abundance of five genes involved in butyric acid synthesis from carbohydrates (*but*, *buk*, *atoA/D* proteins, and *4hbt*) in both the type 2 diabetes and pre-diabetes groups. Interestingly, a flavonoid reductase known as keto reductase was discovered in *F. plautii* American Type Culture Collection (ATCC) 49531, which catalyzes the hydrogenation of flavonoids at the C2=C3 bond, playing a crucial role in the metabolism of flavonoids and influencing the intestinal microbial community (56).

Our comprehensive analyses revealed a significant association between imbalances in the intestinal microbiota and pre-diabetes, particularly noting a decrease in *F. plautii* in both pre-diabetes and diabetes patients. However, the mechanisms of interaction involving *F. plautii* in diabetes prevention and treatment remain limited in understanding.

Moreover, our results indicate that *F. plautii* significantly impacts blood glucose levels, mediated through serum glyceraldehyde. We observed notable positive correlations between several metabolites and blood glucose levels, particularly glyceraldehyde, an essential intermediate in glycolysis. Glyceraldehyde is produced from glucose breakdown via a series of enzymatic reactions and subsequently converted into pyruvate, leading to ATP production. Thus, glyceraldehyde is vital for facilitating energy release from glucose (57). Our data indicated elevated glyceraldehyde levels in the pre-diabetes group, suggesting two possibilities: either an increase in the conversion of glucose to glyceraldehyde or a decrease in the conversion of glyceraldehyde to pyruvate. To elucidate the specific mechanism, further analysis and integration of additional data are warranted. Additionally, other metabolites with increased levels in the pre-diabetes group included L-gulose, gluconolactone, gluconic acid, and galacturonic acid, which likely promote the conversion of glucose to glyceraldehyde while inhibiting its conversion of glyceraldehyde to pyruvate. These findings emphasize abnormalities in the glucose metabolism pathway in the pre-diabetes group, contributing to the reversible elevation in blood glucose levels. Further investigation into the precise mechanisms through which these metabolites affect glucose metabolism and their associations with clinical outcomes is essential. This knowledge could enhance our understanding of pre-diabetes pathogenesis and facilitate the development of targeted interventions aimed at restoring glucose homeostasis.

Notably, *F. plautii* abundance showed a negative correlation with glyceraldehyde levels, which, in turn, positively correlated with blood glucose levels. These findings suggest that *F. plautii* may modulate blood glucose levels by influencing glyceraldehyde metabolism. The implications of these findings extend to the management and treatment of pre-diabetes. By integrating multi-omics data, we can gain deeper insights

into the mechanisms involved in pre-diabetes development, paving new avenues for prevention and treatment.

Additionally, our data revealed a positive correlation between *Bacteroides xylanisolvens* in the gut and blood glucose levels, while *F. plautii* exhibited an inverse relationship. To explore if *F. plautii* influences blood glucose levels through serum glyceraldehyde mediation, we utilized a stepped-mediation analysis model. Results confirmed that *F. plautii* significantly affects blood glucose levels through serum glyceraldehyde (Fig. 5C). Although the negative correlation between *F. plautii* and glucose is not statistically significant (Fig. 5A), our results indicate that *F. plautii* negatively correlates with the intermediate metabolite, glyceraldehyde ($P < 0.05$, $R = -0.31$; Fig. 5D), and glyceraldehyde's positive correlation with blood glucose levels ($P < 0.001$, $R = 0.85$; Fig. 5E) supports the conclusion from Fig. 5C. This reinforces the idea that *F. plautii* may affect blood glucose levels through its impact on glyceraldehyde abundance. However, further larger cohort and animal experiments are needed for validation.

Our focus on *F. plautii* arises from mediation analyses integrating multi-omics data, including gut and oral microbiota, blood metabolites, and blood glucose levels. These analyses revealed *F. plautii*'s potential to influence blood glucose via the metabolite glyceraldehyde (Fig. 5C). Figure 2A, showing the top 15 bacteria species by average relative abundance across three groups, does not include *F. plautii*. This is because *F. plautii*'s abundance is significantly lower in the pre-diabetes and diabetes groups than in the control group (Fig. 2J). This highlights the importance of our innovative multi-omics integration, as the role of *F. plautii* is one of the biologically significant findings that can only be revealed through such comprehensive analyses.

Our comprehensive analysis of multi-omics data provides valuable insights into leveraging the human microbiota and its metabolites for pre-diabetes prevention and treatment. However, given our relatively small sample size, validation of these findings in larger population cohorts is necessary. Furthermore, advancements could be achieved by refining animal models or conducting mechanistic studies, deepening our understanding of these underlying mechanisms and propelling this approach toward new frontiers.

By integrating multiple omics data through comprehensive analysis, we gain a better understanding of the mechanisms underlying pre-diabetes development and remission. This offers new directions and strategies for managing this disease. The intricate relationships among gut microbiota, metabolites, and blood glucose levels emphasized by our research findings are pivotal for exploring personalized treatment methods and preventive strategies for pre-diabetes.

## ACKNOWLEDGMENTS

We are indebted to the staff and participants involved in this study and the GDZYY-cohort 2021. Metagenomic and 16S rRNA gene sequencing analyses were conducted with technical support from Shenzhen KMHD Gene Technology Co., Ltd.

The project was supported by National Key R&D Program of China (2024YFC3506300, 2024YFC3506303), the Specific Fund of State Key Laboratory of Dampness Syndrome of Chinese Medicine (SZ2021ZZ28, SZ2021ZZ3003, and SZ2021ZZ31), Guangzhou Basic and Applied Basic Research Scheme(2024A04J3300), the National Natural Science Foundation of China (No. 82004234), the Science and Technology Planning Project of Guangdong Province (2017B030314166 and 2020B1111100005), the Foundation of Guangdong Provincial Hospital of Traditional Chinese Medicine (YN2020ZWB04), Innovation Team and Talents Cultivation Program of National Administration of Traditional Chinese Medicine (ZYYCXTD-C-202001), and Guangzhou Science and Technology Bureau City School Joint Project (2023 A03J0754).

Y.M.L. and L.H. conceived and devised the study. Q.W.Q., Y.S.D., X.X.S., C.R.W., and C.S. were responsible for the acquisition of data. Y.M.L., S.Y.C., L.H., W.H., H.W., Y.C., and Z.M.Y. analyzed the data. Y.M.L. drafted the article with contributions from Y.C. and Z.M.Y.

Y.M.L. and L.H. are the guarantors of this work. All authors contributed to the interpretation of the data, reviewed and edited the manuscript, and approved the version to be published.

## AUTHOR AFFILIATIONS

[1]State Key Laboratory of Traditional Chinese Medicine Syndrome, State Key Laboratory of Dampness Syndrome of Chinese Medicine Syndrome, The Second Affiliated Hospital of Guangzhou University of Chinese Medicine, Guangzhou, China
[2]KMHD, Shenzhen, China
[3]Department of Endocrinology and Metabolism, The Fifth Affiliated Hospital of Sun Yat-sen University, Zhuhai, China

## AUTHOR ORCIDs

Yanmin Liu  http://orcid.org/0000-0003-3731-0865
Xiaodong Fang  http://orcid.org/0000-0001-7061-3337
Li Huang  http://orcid.org/0000-0002-9087-6342

## FUNDING

| Funder | Grant(s) | Author(s) |
| --- | --- | --- |
| National Key R&D Program of China | 2024YFC3506300, 2024YFC3506303 | Li Huang |

## AUTHOR CONTRIBUTIONS

Yanmin Liu, Conceptualization, Formal analysis, Writing – original draft, Writing – review and editing | Qinwei Qiu, Formal analysis, Methodology | Yang Chen, Formal analysis, Project administration, Supervision | Yusheng Deng, Formal analysis, Methodology, Writing – original draft, Writing – review and editing | Wei Huang, Funding acquisition, Project administration | Chen Sun, Funding acquisition, Investigation, Resources | Xiaoxiao Shang, data curation, Formal analysis, Project administration | Xinyan Chen, data curation, validation, Writing – original draft | Chengrui Wang, data curation, Formal analysis, Resources | Lijuan Han, Investigation, Methodology, Project administration, Resources | Shiyan Chen, Formal analysis, Writing – review and editing | Jiamin Yuan, data curation, Resources | Fuping Xu, data curation, Resources | Zhimin Yang, Investigation, Supervision | Xiaodong Fang, Supervision, Writing – review and editing | Li Huang, Conceptualization, Investigation, Project administration, Resources, Writing – review and editing

## DATA AVAILABILITY

The metagenome and 16S rRNA sequencing data from the patients and controls are publicly accessible through the National Genomics Data Center GSA repository via the links https://bigd.big.ac.cn/gsa/browse/CRA025712 and https://bigd.big.ac.cn/gsa-human/browse/HRA011514.

## ETHICS APPROVAL

This study was approved by the Ethics Committee of Guangdong Provincial Hospital of Chinese Medicine (Ethical review number: B2017-199-02). All participants were from Guangdong Provincial Hospital of Chinese Medicine, Fangcun Hospital, and gave their written informed consent to publish their clinical data, including all images, clinical data, and other data included in this article.

## ADDITIONAL FILES

The following material is available online.

### Supplemental Material

**Fig. S1 (Spectrum01459-24-s0001.pdf).** Metabolic profile alterations among the Diabetes group, Healthy control group, and Pre-diabetes group.
**Fig. S2 (Spectrum01459-24-s0002.pdf).** Metabolic alterations in glucose pathway distinguish Pre-diabetes from Control groups without outlier sample.
**Supplemental legends (Spectrum01459-24-s0003.docx).** Legends for Fig. S1 and S2.

### Open Peer Review

**PEER REVIEW HISTORY (review-history.pdf).** An accounting of the reviewer comments and feedback.

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
