## [Reviewer comments · Microbiology Spectrum]

Microbiology Spectrum

Integrated Multi-omics Analysis Reveals the Functional Signature of Microbes and Metabolomics in Pre-diabetes individuals

Yanmin Liu, Qinwei Qiu, Yang Chen, Yusheng Deng, Wei Huang, Chen Sun, Xiaoxiao Shang, Xinyan Chen, Chengrui Wang, Lijuan Han, Shiyang Chen, Jiamin Yuan, Fuping Xu, Zhimin Yang, Xiaodong Fang, and Li Huang

Corresponding Author(s): Yanmin Liu, The Second Clinical Medical College, Guangzhou University of Chinese Medicine

Review Timeline:

Submission Date:	June 13, 2024
Editorial Decision:	October 21, 2024
Revision Received:	November 20, 2024
Editorial Decision:	January 30, 2025
Revision Received:	April 2, 2025
Accepted:	April 7, 2025

Editor: Se-Ran Jun

Reviewer(s): Disclosure of reviewer identity is with reference to reviewer comments included in decision letter(s). The following individuals involved in review of your submission have agreed to reveal their identity: Leo Gerlin (Reviewer #1)

Transaction Report:

DOI: <https://doi.org/10.1128/spectrum.01459-24>

Re: Spectrum01459-24 (Integrated Multi-omics Analysis Reveals the Functional Signature of Microbes and Metabolomics in Pre-diabetes individuals)

Dear Dr. Li Huang:

Thank you for the privilege of reviewing your work. Below you will find my comments, instructions from the Spectrum editorial office, and the reviewer comments.

Revision Guidelines

Sincerely,
Se-Ran Jun
Editor
Microbiology Spectrum

Reviewer #1 (Comments for the Author):

Comments and Suggestions for the Author:

Liu et al. analyzed gut and tongue microbiomes and serum metabolomes for three groups: healthy controls, prediabetic patients and diabetic patients. They use statistical modeling to build links between all the acquired data. Altogether, their work helps

deciphering novel signatures of prediabetic stage and understand the processes (physiology, metabolism, microbiota) associated with the development of the disease. It constitutes a valuable dataset on this disease and provide interesting results on the specific actions of some bacteria and associated metabolites. However, numerous parts of the result section require more precision or references. In addition, some statements must be corrected as they are not in line with what is shown in the figures. Once these points are corrected, this paper will constitute a valuable study for the community.

The division of the sections into "Result 1" "Result 2"... is strange and might not be necessary. The section title itself is sufficient. Overall, the section Results must be more concise and focus on describing the findings. Discussion must also be more concise and focus on explaining and making links based on the results obtained in the study. Digressions must be removed from the two sections.

I. 117 - 118: "intestinal mucin proteins (...) were identified in fecal samples, which is consistent with the relative abundance of Akkermansia muciniphila"

The link between mucin proteins and *A. muciniphila* is unclear here, it should be explicated or removed.

I. 137: it should be "the gut bacterium" instead of "the gut microbiota"

Figure 1: units are not mentioned. They should appear in the figure or in the legend.

I. 234 "glucose levels" and I. 237 "glutamyl transpeptidase levels" is unclear. "Concentration" might be more accurate.

I. 238: the expression "suggesting potential liver dysfunction" is very speculative and is more accurate in the Discussion, or should be rephrased if it is a well-established fact.

I. 240 - 241: the authors claim that "the liver dysfunction is prior to the blood glucose levels". Figure 1 data alone do not prove that. The two markers (GGT and glucose) increase between healthy and PD states. The fact that the increase is then not significant between PD and diabetic stages for GGT is not enough to prove that GGT (related to liver dysfunction) occurs first. We could imagine that both occur simultaneously at the transition from healthy to PD, and then that liver dysfunction plays a more limited role between PD to diabetic stages.

I. 246 - 253: all this part lacks bibliographical references, and would better fit in Discussion section

I. 270 - 274 and 279-280: Bibliographical references missing.

I. 293 - 294, I. 303: This is a wrong statement as increase between normal and PD is not significant.

I.301: This is a wrong statement as decrease between normal and PD is not significant.

I. 303: "exhibiting various metabolic capabilities" It is not proved here with your data. Should be moved to Discussion or removed.

I. 326 - 328: Bibliographical references missing.

I. 333 - 339 : Bibliographical references missing.

I. 354 - 357 : The authors mention glyceraldehyde as "the key metabolite". Whether it is true that it has the highest contribution, several other metabolites are not even commented while they have a very close score and are also key contributors to the discrimination (methylthioadenosine, taurine, benzene triol). This clearly means that glyceraldehyde alone is not sufficient to discriminate the groups.

I. 386: "regulatory effects on glucose metabolism" Bibliographical reference missing.

I. 389 - 390: "the differential metabolite profiles (...) primarily centered around alterations in the glucose metabolism" Maybe not only if we look at the other metabolites with high contribution to the discrimination such as methylthioadenosine, taurine, benzene triol.

I. 415: it is unclear why GGT has been pragmatically chosen here and not another markers.

I. 475: should be "is higher in" instead of "in higher in"

I. 524: should be "Using" instead of "Usin21g"

The statistical tests are not mentioned in the figure legends, in particular Figure 1 and Figure 4C. (it is briefly explained in the

Methods for gut and tong microbiota)

Figure 4C: concentration unit is not mentioned

Reviewer #1 (Comments for the Author):

Comments and Suggestions for the Author:

Liu et al. analyzed gut and tongue microbiomes and serum metabolomes for three groups: healthy controls, prediabetic patients and diabetic patients. They use statistical modeling to build links between all the acquired data. Altogether, their work helps deciphering novel signatures of prediabetic stage and understand the processes (physiology, metabolism, microbiota) associated with the development of the disease. It constitutes a valuable dataset on this disease and provide interesting results on the specific actions of some bacteria and associated metabolites. However, numerous parts of the result section require more precision or references. In addition, some statements must be corrected as they are not in line with what is shown in the figures. Once these points are corrected, this paper will constitute a valuable study for the community.

The division of the sections into "Result 1" "Result 2"... is strange and might not be necessary. The section title itself is sufficient. Overall, the section Results must be more concise and focus on describing the findings. Discussion must also be more concise and focus on explaining and making links based on the results obtained in the study. Digressions must be removed from the two sections.

We appreciate the reviewer's feedback regarding the organization and conciseness of the Results and Discussion sections.

Section Titles: We agree that dividing the Results into "Result 1," "Result 2," etc., may be unnecessary and have removed these labels in the revised manuscript.

Conciseness: We acknowledge the need for greater conciseness in both the Results and Discussion sections. We have thoroughly revised these sections to emphasize the key findings and their implications, ensuring that extraneous information and digressions have been removed. Our objective is to present a clearer and more focused narrative that directly connects our results to the broader context of the study.

We believe these changes enhance the clarity and impact of the manuscript and appreciate the reviewer's guidance in improving the overall quality of our work.

Thank you for your valuable suggestions.

I. 117 - 118: "intestinal mucin proteins (...) were identified in fecal samples, which is consistent with the relative abundance of Akkermansia muciniphila"

The link between mucin proteins and A. muciniphila is unclear here, it should be explicated or removed.

Thank you for your careful review and constructive feedback on our manuscript. We appreciate your observation regarding the clarity of the link between mucin proteins and *Akkermansia muciniphila*.

To address this concern, we have revised the relevant section to explicitly clarify the relationship between intestinal mucin proteins (MUC-1 and MUC-2) and *Akkermansia muciniphila*. We will elaborate on how *A. muciniphila* utilizes these mucins as a primary carbon source and highlight its ability to degrade them for nutrient acquisition, which is crucial for its growth and reproduction. (Line 106-112)

We believe that this connection is important for understanding the ecological role of *A. muciniphila* within the gut microbiome, and we are committed to making this clearer for readers.

Thank you once again for your valuable suggestion!

I. 137: it should be "the gut bacterium" instead of "the gut microbiota"

Thank you for your insightful feedback regarding the terminology used in our manuscript. We appreciate your suggestion to clarify the distinction between "the gut microbiota" and "the gut bacterium."(Line 131)

Figure 1: units are not mentioned. They should appear in the figure or in the legend.

Thank you for pointing out the omission of units in Figure 1. We appreciate your attention to detail. In response, we have added the appropriate units either directly within the figure to ensure that the information is clear and accessible to readers.

I. 234 "glucose levels" and I. 237 "glutamyl transpeptidase levels" is unclear. "Concentration" might be more accurate.

Thank you for your valuable feedback. We recognize the importance of using precise terminology in scientific and medical contexts. Accordingly, we have revised "glucose levels" to "glucose concentration" (Line 228) and updated "glutamyl transpeptidase levels" to "glutamyl transpeptidase concentration" (Line 231).

I. 238: the expression "suggesting potential liver dysfunction" is very speculative and is more accurate in the Discussion, or should be rephrased if it is a well-established fact.

Thank you for your valuable feedback regarding Line 238. We have revised the statement to reflect a stronger association between elevated GGT levels and liver dysfunction. The updated text now reads: "Additionally, serum glutamyl transpeptidase (GGT) concentration showed a significant increase in both the Pre-diabetes group and Diabetes group ($P < 0.05$) (Figure 1B), which is indicative of liver dysfunction." (Line 230-232)

Thank you for your feedback. This change aims to convey a more definitive connection based on established research.

I. 240 - 241: the authors claim that "the liver dysfunction is prior to the blood glucose levels". Figure 1 data alone do not prove that. The two markers (GGT and glucose) increase between healthy and PD states. The fact that the increase is then not significant between PD and diabetic stages for GGT is not enough to prove that GGT (related to liver dysfunction) occurs first. We could imagine that both occur simultaneously at the transition from healthy to PD, and then that liver dysfunction plays a more limited role between PD to diabetic stages.

Thank you for your thoughtful critique regarding our statement that "liver dysfunction is prior to the blood glucose levels." We appreciate your insights and recognize the need for clarity regarding the interpretation of the data.

To address your concern, we have revised our explanation to enhance its clarity and precision. The updated text now reads: "Additionally, serum glutamyl transpeptidase (GGT) concentration were significantly increased in both the Pre-diabetes and Diabetes groups ($P < 0.05$) (Figure 1B), which is indicative of liver dysfunction^{17,18}. Interestingly, our results demonstrate that GGT concentrations are significantly elevated in the pre-diabetes (PD) stage compared to the normal stage, with no further significant increase observed between the PD and diabetes stages. In contrast, blood glucose concentration shows a continuous rise from the normal stage through the PD stage and into the diabetes stage. This pattern implies that hepatic dysfunction, as indicated by the elevated GGT concentrations, is established during the PD stage and may precede the subsequent increase in blood glucose levels. These findings highlight the potential for early liver involvement in metabolic dysregulation and support the notion that alterations in hepatic function play a critical role in the pathogenesis of glucose metabolism disorders. Furthermore, they underscore the potential utility of GGT as a valuable biomarker for identifying individuals at risk of developing diabetes." (Line 230-245)

The revised passage emphasizes the need for further investigation to elucidate the dynamics between liver function and glucose homeostasis during these metabolic transitions.

Thank you again for your valuable feedback, which has helped enhance the clarity and precision of our manuscript.

I. 246 - 253: all this part lacks bibliographical references, and would better fit in Discussion section

We appreciate your valuable feedback and have added appropriate citations to support the statements made in this part of the manuscript and moved the relevant text accordingly and ensured that it is well-integrated with existing discussions (Line 455-459).

Thank you for helping us improve the clarity and rigor of our manuscript.

I. 270 - 274 and 279-280: Bibliographical references missing.

Thank you for your careful review. We acknowledge that bibliographical references were missing and have added the appropriate citations to these sections. (Line 270, 271,276)

I. 293 - 294, I. 303: This is a wrong statement as increase between normal and PD is not significant.

Thank you for pointing out this inconsistency. We have updated the statement accordingly to reflect the correct observation: “Additionally, we noted an increase in the abundance of *Bacteroides xylanisolvens* in the Diabetes groups” (Line 297 and Line 307)

I.301: This is a wrong statement as decrease between normal and PD is not significant.

Thank you for your feedback regarding line 301, as follow “The abundance of *F. plautii* was negatively correlated with blood glucose levels, suggesting its potential involvement in blood glucose regulation through butyric acid production.”. We would like to clarify that we did not find a similar expression or statement in the original manuscript at this line. It is possible that there has been a misunderstanding or miscommunication regarding this point. If the reviewer could provide more specific details about the statement in question, we would be happy to address it appropriately.

I. 303: "exhibiting various metabolic capabilities" It is not proved here with your data. Should be moved to Discussion or removed.

Thank you for your insightful feedback. We have removed it entirely to ensure that our conclusions are firmly supported by the data. (Line 307)

I. 326 - 328: Bibliographical references missing.

Thank you for your thorough review. We have added the appropriate citations to these lines to ensure that all statements are properly supported by the literature. (Line 329)

I. 333 - 339: Bibliographical references missing.

Thank you for your thorough review. We have added the appropriate citations to these lines to ensure that all statements are properly supported by the literature. (Line 337)

I. 354 - 357: The authors mention glyceraldehyde as "the key metabolite". Whether it is true that it has the highest contribution, several other metabolites are not even commented while they have a very close score and are also key contributors to the discrimination (methylthioadenosine, taurine, benzene triol). This clearly means that glyceraldehyde alone is not sufficient to discriminate the groups.

Thank you for your valuable feedback regarding our characterization of glyceraldehyde as “the key metabolite” in lines 354-357. We appreciate your point that while glyceraldehyde may contribute significantly to the discrimination between groups, it is not the sole metabolite responsible for this effect. We have revised the manuscript to ensure a more balanced representation of these important metabolites and clarify that glyceraldehyde. “The key metabolites that contributing most to the discrimination between the Pre-diabetes group and

Normal group include Glyceraldehyde, L-Gulose, N-methyl L-glutamic Acid, benzene triol, Hydroxymethylglutaric acid, Taurine, and Methylthioadenosine (Figure 4B)" (Line 351-354)

I. 386: "regulatory effects on glucose metabolism" Bibliographical reference missing.
Thank you for your thorough review. We have added the appropriate citations to these lines to ensure that all statements are properly supported by the literature. (Line 383)

I. 389 - 390: "the differential metabolite profiles (...) primarily centered around alterations in the glucose metabolism" Maybe not only if we look at the other metabolites with high contribution to the discrimination such as methylthioadenosine, taurine, benzene triol.

Thank you for your insightful feedback regarding lines 389-390. We appreciate the reviewer's valuable observation that glyceraldehyde is not the sole differential metabolite between the Pre-diabetes and Normal groups. We fully agree with this perspective, and to provide a more accurate representation, we have updated the relevant descriptions in the manuscript (Lines 385-386) to reflect the contributions of other significant metabolites involved in the discrimination between these groups.

I. 415: it is unclear why GGT has been pragmatically chosen here and not another marker.

Thank you for your insightful feedback regarding our choice of GGT as a marker in line 415. We selected GGT (gamma-glutamyl transferase) primarily because it is one of the significant clinical indicators that shows marked changes alongside glucose (GLU) levels. Our analysis reveals that GGT levels are significantly elevated in both the PD group and the diabetic group compared to the Normal group (Figure 1B).

Additionally, during our correlation analysis, we found that GGT exhibited the strongest associations with other omics data. Finally, it is important to note that GGT plays a critical role in glucose metabolism, further justifying its selection as a relevant marker in our study.

We will incorporate these detailed explanations into the manuscript to enhance clarity and provide a stronger rationale for our choice of GGT. (Line 411-417)

I. 475: should be "is higher in" instead of "in higher in"

Thank you for your careful review. We update "in higher in" be corrected to "is higher in". (Line 479)

I. 524: should be "Using" instead of "Usin21g"

Thank you for pointing out the typographical error. We have corrected the revised manuscript as "To further verified that *F. plautii* influences blood glucose levels through serum glyceraldehyde mediation, we utilized a stepped-mediation analysis model, confirming that *F. plautii* significantly affects blood glucose levels through serum glyceraldehyde". (Line 523)

The statistical tests are not mentioned in the figure legends, in particular Figure 1 and Figure 4C. (it is briefly explained in the Methods for gut and tong microbiota)

Thank you for your valuable feedback. We have now added the relevant statistical information to the figure legends for both figures (Line 745-747 and Line 785-786) and supplement in the method (Line 185-186).

Figure 4C: concentration unit is not mentioned

Thank you for your valuable feedback regarding Figure 4C. To avoid any misunderstanding, we will update the figure legend to replace "concentration" with " Intensity," as the data presented represents the density of non-targeted metabolites and does not have specific concentration units.

We appreciate your attention to detail and believe this adjustment will enhance the clarity of our manuscript.

Re: Spectrum01459-24R1 (Integrated Multi-omics Analysis Reveals the Functional Signature of Microbes and Metabolomics in Pre-diabetes individuals)

Dear Dr. Yanmin Liu:

Thank you for the privilege of reviewing your work. Below you will find my comments, instructions from the Spectrum editorial office, and the reviewer comments.

I have received comments from the two reviewers that must be addressed. Especially, the second reviewer raised major concerns, about sample size.

Revision Guidelines

Sincerely,
Se-Ran Jun
Editor
Microbiology Spectrum

Reviewer #1 (Comments for the Author):

The revised version of the manuscript by Liu et al. has improved the phrasing and conciseness, which I acknowledge. I appreciated reading this version, which provided a clearer insight into the work done and the main results and conclusions. However, some parts of the text, particularly the analysis of the figures and the associated conclusions, remain confusing and

sometimes misleading.

L. 87 - 88 : "it is estimated that by 2040, the number of individuals affected by PD will exceed million."

None of the bibliographical references put in this paragraph seems to support this fact.

L.106 - 108 : "Therefore, gaining a comprehensive understanding of the mechanisms underlying the human microbiota through the integration of multi-omics approaches is of vital importance." And L. 129 - 131 : "Elucidating the mechanisms underlying the host-microbial interaction in the pathogenesis of pre-diabetes (PD) through the integration of multi-omics data is of immense significance."

The two sentences are almost the same, put at different places of the Introduction. Some conciseness is still required here.

L. 250 - 252 : "Furthermore, both the Pre-diabetes and Diabetes groups exhibited a significant reduction in high-density lipoprotein (HDL) levels (Figure 251 1D, $P < 0.05$)."

The statistical tests, as presented in Figure 1D, show no significant difference between normal and pre-diabetes levels.

L. 281- 290 : "For example" to "blood glucose levels". + L. 297 - 305 : "Butyric acid" to "prediabetes."

These parts should be moved to Discussion.

L. 325 - 327 : "Of particular interest, the abundance of *Corynebacterium* was significantly lower in the Pre-diabetes group compared to both the Normal and the Diabetes groups (Figure 3A)."

This is in Figure 3B and not 3A.

L. 329 : "Thses »

Typo

Figure 4A

One point (blue, in the top right corner) has a very distinct metabolic profile compared to all others. It could heavily impact the other statistical analysis and provide false positives. The authors have to provide an explanation to this distinct profile and analyze if it affects statistical tests in Figure 4B and 4C.

L. 389 - 390 : "our findings revealed that N-methyl-L-glutamic acid (Figure 4C), another significant metabolite, exerts regulatory effects on glucose metabolism"

The findings in the study did not revealed regulatory effects as they are only correlations. This statement should thus be reconsidered.

L. 431 - 432 : "Our analysis provided compelling evidence for the significant role of *F. plautii* in modulating blood glucose levels"

Could you explain why *F. plautii* appears such great interest, while in Figure 5A, all the associations with *F. plautii* are not significant associations? I am also puzzled that it is absent from Figure 2A.

Reviewer #3 (Public repository details (Required)):

Metabolomics and Microbiome data

Reviewer #3 (Comments for the Author):

Spectrum01459-24R1

Liu et al.

Summary and Critiques

The investigators explore the potential shifts in microbiota communities (tongue and stool) plus serum metabolome in healthy adults, subjects with pre-diabetes, and subjects with diagnosed T2DM. Emerging science suggests that even in the pre-DM state, changes in gut microbiota communities and functions are altered, so new information and clarity in this arena can be valuable to begin to consider early events in pathogenesis and risk. Despite the importance of this area, the current experiments

are severely underpowered, statistics are not proper for the questions being asked, and the authors overinterpret the limited results. These present fatal flaws to the paper.

Comments:

(1) There are no power calculations presented or alternatively, a premise provided for the subject numbers per group. For "omics" type studies in which correlation statistics and multivariate models are to be applied, greater confidence is needed that the statistical approaches will avoid Type 1 and Type 2 errors. This is not the case here, where too few subjects in the target group (pre-DM) are evident. Furthermore, use of t-tests is not appropriate when considering multiple groups, giving this reviewer limited confidence in all of the statistical outcomes presented in the paper. In other places, comparisons are only made between healthy and Pre-DM and no T2DM are presented (i.e., PLS-DA model and related figures); this is without explanation.

(2) There are many examples where the authors use words and phrases suggestive of "action" or "cause" even though all that is presented is some type of association. Sometimes, the latter are not even backed up by formal statistics but simply two variables that go in the same direction or opposite direction. Furthermore, even when correlations are applied, as in Figure 5, the conclusions are dubious-for instance, the authors refer to a negative correlation between *F. plautii* and glucose, yet this is not statistically significant in the figure and it is not clear from the figure legend which subject data were used for the correlation matrix (all participants?). In Figure 5D, the *F. plautii* correlation with glyceraldehyde is essentially flat and yet it is implied to be important and by extension critical to glucose metabolism. This and many other examples indicate that the investigators are overinterpreting the results to be physiologically relevant when they may not be.

This critique extends to the abstract and paper summary, which suggest the results somehow are applicable to personalized interventions and management of diabetes.

(3) Miscellaneous: It is unclear why so few participants' data were included in the study since the cohort being examined is >300 persons. Did the study adhere to the Declaration of Helsinki parameters? Please use "sex" not "gender" since sex is the proper scientific term in this context. Instrument settings, analysis details, and sample preparation details are lacking for the metabolomics methods.

The revised version of the manuscript by Liu et al. has improved the phrasing and conciseness, which I acknowledge. I appreciated reading this version, which provided a clearer insight into the work done and the main results and conclusions. However, some parts of the text, particularly the analysis of the figures and the associated conclusions, remain confusing and sometimes misleading.

L. 87 – 88 : “it is estimated that by 2040, the number of individuals affected by PD will exceed million.”

None of the bibliographical references put in this paragraph seems to support this fact.

L.106 – 108 : “Therefore, gaining a comprehensive understanding of the mechanisms underlying the human microbiota through the integration of multi-omics approaches is of vital importance.” And L. 129 – 131 : “Elucidating the mechanisms underlying the host-microbial interaction in the pathogenesis of pre-diabetes (PD) through the integration of multi-omics data is of immense significance.”

The two sentences are almost the same, put at different places of the Introduction. Some conciseness is still required here.

L. 250 – 252 : “Furthermore, both the Pre-diabetes and Diabetes groups exhibited a significant reduction in high-density lipoprotein (HDL) levels (Figure 251 1D, $P < 0.05$).”

The statistical tests, as presented in Figure 1D, show no significant difference between normal and pre-diabetes levels.

L. 281– 290 : “For example” to “blood glucose levels”. + L. 297 – 305 : “Butyric acid” to “prediabetes.”

These parts should be moved to Discussion.

L. 325 – 327 : “Of particular interest, the abundance of *Corynebacterium* was significantly lower in the Pre-diabetes group compared to both the Normal and the Diabetes groups (Figure 3A).”

This is in Figure 3B and not 3A.

L. 329 : “Thses »

Typo

Figure 4A

One point (blue, in the top right corner) has a very distinct metabolic profile compared to all others. It could heavily impact the other statistical analysis and provide false positives. The authors have to provide an explanation to this distinct profile and analyze if it affects statistical tests in Figure 4B and 4C.

L. 389 – 390 : “our findings revealed that N-methyl-L-glutamic acid(Figure 4C), another significant metabolite, exerts regulatory effects on glucose metabolism”

The findings in the study did not revealed regulatory effects as they are only correlations. This statement should thus be reconsidered.

L. 431 – 432 : “Our analysis provided compelling evidence for the significant role of *F. plautii* in modulating blood glucose levels”

Could you explain why *F. plautii* appears such great interest, while in Figure 5A, all the associations with *F. plautii* are not significant associations? I am also puzzled that it is absent from Figure 2A.

Reviewer #1 (Comments for the Author):

The revised version of the manuscript by Liu et al. has improved the phrasing and conciseness, which I acknowledge. I appreciated reading this version, which provided a clearer insight into the work done and the main results and conclusions. However, some parts of the text, particularly the analysis of the figures and the associated conclusions, remain confusing and sometimes misleading.

L. 87 - 88: "it is estimated that by 2040, the number of individuals affected by PD will exceed million."

None of the bibliographical references put in this paragraph seems to support this fact.

We sincerely appreciate the reviewer's attention to the lack of reference support for our estimate. In response, we have incorporated references from JAMA 2023 and Diabetes Research and Clinical Practice 2022. These references offer a comprehensive analysis of global prediabetes prevalence, thereby strengthening our estimate mentioned (Line97).

L.106 - 108: "Therefore, gaining a comprehensive understanding of the mechanisms underlying the human microbiota through the integration of multi-omics approaches is of vital importance." And L. 129 - 131: "Elucidating the mechanisms underlying the host-microbial interaction in the pathogenesis of pre-diabetes (PD) through the integration of multi-omics data is of immense significance."

The two sentences are almost the same, put at different places of the Introduction. Some conciseness is still required here.

Following the reviewer's suggestion, we have carefully refined the text to eliminate redundancy and improve clarity. The sentence, "Therefore, gaining a comprehensive ... is of vital importance.", has been deleted for it was deemed redundant (Line113-115). And "Elucidating the mechanisms ... is of immense significance." has been rephrased to "Elucidating the mechanisms of host - microbial interaction in pre - diabetes (PD) pathogenesis by integrating multi - omics data is highly significant." to make it more concise and precise (Line133-134).

L. 250 - 252: "Furthermore, both the Pre-diabetes and Diabetes groups exhibited a significant reduction in high-density lipoprotein (HDL) levels (Figure 251 1D, P< 0.05)."

The statistical tests, as presented in Figure 1D, show no significant difference between normal and pre-diabetes levels.

We appreciate your attention to detail. We have revised the description to align with the actual statistical results. "Compared to the Normal group, the Diabetes group showed a significant reduction in HDLC levels (Figure 1D, $P < 0.05$). Although the Pre - diabetes group's HDLC levels were similar to the Diabetes group's and decreased noticeably from the Normal group's, this difference was not statistically significant ((Figure 1D)." (Line267-269)

L. 281- 290: "For example" to "blood glucose levels". + L. 297 - 305: "Butyric acid" to "prediabetes."

These parts should be moved to Discussion.

Following the reviewer's suggestion, we have moved the specified content to the Discussion section for better structural coherence (Line492-505).

L. 325 - 327: "Of particular interest, the abundance of Corynebacterium was significantly lower in the Pre-diabetes group compared to both the Normal and the Diabetes groups (Figure 3A)."

This is in Figure 3B and not 3A.

We have fixed the figure label error as per your reminder (Line330). Additionally, we have double-checked all figure labels and references in the text to ensure consistency and accuracy.

L. 329: "Thses »

Typo

The typographical error has been corrected (Line332). We have also performed a final proofread to ensure the text is free of typos and formatting issues.

Figure 4A

One point (blue, in the top right corner) has a very distinct metabolic profile compared to all others. It could heavily impact the other statistical analysis and

provide false positives. The authors have to provide an explanation to this distinct profile and analyze if it affects statistical tests in Figure 4B and 4C.

We sincerely appreciate the reviewer's insightful observation regarding the outlier sample in the normal group. Following the reviewer's suggestion, we have conducted the relevant analyses and incorporated into the manuscript accordingly: "It is noteworthy that there is an outlier in the normal group. To assess its impact on the overall analysis, we first analyzed the metabolic signature of the outlier is characterized by significantly reduced levels of specific metabolites, including N-Acetyl-glutamine, L-Tyrosine, and Pyrrolidine. Subsequently, we reperformed statistical analyses excluding this sample. As shown in Figure supplement 1, the overall trends remained largely consistent. Glyceraldehyde continued to be the primary differentiator between the two groups. The order of contribution from other metabolites and the abundance of differential metabolites in Figure 4C also stayed the same. Moreover, this result confirmed that including this outlier did not alter our fundamental conclusions." (Line402-411)

L. 389 - 390: "our findings revealed that N-methyl-L-glutamic acid (Figure 4C), another significant metabolite, exerts regulatory effects on glucose metabolism"

The findings in the study did not reveal regulatory effects as they are only correlations. This statement should thus be reconsidered.

We have refined the conclusion to precisely mirror the correlational essence of the results and ensure the inclusion of the regulatory effects noted in the references. "Furthermore, our findings indicate an increase in N-methyl-L-glutamic acid in the Pre-diabetes group (Figure 4C). Prior studies have demonstrated that this metabolite exerts regulatory effects on glucose metabolism." (Line398-400)

L. 431 - 432: "Our analysis provided compelling evidence for the significant role of *F. plautii* in modulating blood glucose levels"

Could you explain why *F. plautii* appears such great interest, while in Figure 5A, all the associations with *F. plautii* are not significant associations? I am also puzzled that it is absent from Figure 2A.

We appreciate the reviewer's insightful question regarding the emphasis on *F. plautii* and have added a brief discussion on why *F. plautii*'s abundance is lower in the Pre-diabetes and Diabetes groups and how this relates to its potential impact on blood

glucose levels: “Our focus on *F. plautii* arises from mediation analyses integrating multi-omics data, including gut and oral microbiota, blood metabolites, and blood glucose levels. These analyses revealed *F. plautii*'s potential to influence blood glucose via the metabolite Glyceraldehyde (Figure 5C). Figure 2A, showing the top 15 bacteria species by average relative abundance across three groups, does not include *F. plautii*. This is because *F. plautii*'s abundance is significantly lower in the Pre - diabetes and Diabetes groups than in the Normal group (Figure 2J). This highlights the importance of our innovative multi-omics integration, as the role of *F. plautii* is one of the biologically significant findings that can only be revealed through such comprehensive analyses.” (Line573-581)

Reviewer #3 (Public repository details (Required)):

Metabolomics and Microbiome data

Reviewer #3 (Comments for the Author):

Spectrum01459-24R1

Liu et al.

Summary and Critiques

The investigators explore the potential shifts in microbiota communities (tongue and stool) plus serum metabolome in healthy adults, subjects with pre-diabetes, and subjects with diagnosed T2DM. Emerging science suggests that even in the pre-DM state, changes in gut microbiota communities and functions are altered, so new information and clarity in this arena can be valuable to begin to consider early events in pathogenesis and risk. Despite the importance of this area, the current experiments are severely underpowered, statistics are not proper for the questions being asked, and the authors overinterpret the limited results. These present fatal flaws to the paper.

Comments:

(1) There are no power calculations presented or alternatively, a premise provided for the subject numbers per group. For "omics" type studies in which correlation statistics and multivariate models are to be applied, greater confidence is needed that the statistical approaches will avoid Type 1 and Type 2 errors. This is not the case here, where too few subjects in the target group (pre-DM) are evident. Furthermore, use of t-tests is not appropriate when considering multiple groups, giving this reviewer limited confidence in all of the statistical outcomes presented in the paper. In other places, comparisons are only made between healthy and Pre-DM and no T2DM are presented (i.e., PLS-DA model and related figures); this is without explanation.

Power Calculation and Sample Size:

We appreciate the reviewer's comment regarding power calculations. While a priori power calculations are indeed a methodological gold standard in prospective studies with clear hypotheses and endpoints, applying them to exploratory, cross-sectional "omics" studies like ours presents is challenging.

Our cross-sectional study examines omics' profiles across individuals at varying glucose tolerance stages. We initially recruited over 300 participants, grouping them by glycemic criteria. Focusing on human microbiota and blood-glucose-related metabolites, we matched age and gender across groups to control for influencing factors. This resulted in 31 healthy controls, 10 T2DM individuals, and 7 pre-diabetes (pre-DM) individuals.

We recognize our relatively small sample size, especially in the pre-DM group. To address this, we've added a discussion section outlining these constraints: "Although our comprehensive analysis of multi-omics data provides valuable insights into leveraging the human microbiota and its metabolites for prediabetes prevention and treatment. However, given our relatively small sample size, validation of these findings in larger population cohorts is necessary. Furthermore, advancements could be achieved by refining animal models or conducting mechanistic studies, deepening our understanding of these underlying mechanisms and propelling this approach towards new frontiers." (Line582-588).

Appropriateness of Statistical Tests:

We appreciate the reviewer's suggestion regarding the statistical approach. We have revised the analysis to use the Wilcoxon test, which is more suitable for our data given the non-parametric nature of the metabolite abundance measurements (Line224). Moreover, we have incorporated a detailed explanation of the omics data analysis method. "For the multivariate comparisons in microbiome analysis, we performed permutational multivariate analysis of variance (PERMANOVA) tests, and analyzed

the metabolome using Partial Least Squares Discriminant Analysis (PLS-DA).”
(Line194-197)

Inclusion of T2DM in PLS-DA and Related Figures:

We have incorporated comparative data results between the diabetes group and the healthy group, as well as between the diabetes group and the pre-diabetes group using PLS-DA. This inclusion enhances the understanding of the disease progression and provides a more comprehensive view of the metabolic changes. (Line360-368)

(2) There are many examples where the authors use words and phrases suggestive of "action" or "cause" even though all that is presented is some type of association. Sometimes, the latter are not even backed up by formal statistics but simply two variables that go in the same direction or opposite direction. Furthermore, even when correlations are applied, as in Figure 5, the conclusions are dubious—for instance, the authors refer to a negative correlation between *F. plautii* and glucose, yet this is not statistically significant in the figure and it is not clear from the figure legend which subject data were used for the correlation matrix (all participants?). In Figure 5D, the *F. plautii* correlation with glyceraldehyde is essentially flat and yet it is implied to be important and by extension critical to glucose metabolism. This and many other examples indicate that the investigators are overinterpreting the results to be physiologically relevant when they may not be.

This critique extends to the abstract and paper summary, which suggest the results somehow are applicable to personalized interventions and management of diabetes.

We sincerely appreciate the reviewer's insights comments on emphasizing causal relationships in our statements. In response, we have added a section in the Discussion to elaborate on the implications of our findings. Specifically, “To explore if *F. plautii* influences blood glucose levels through serum glyceraldehyde mediation, we utilized a stepped-mediation analysis model. Results confirmed that *F. plautii* significantly affects blood glucose levels through serum glyceraldehyde (Figure 5C). Although the negative correlation between *F. plautii* and glucose isn't statistically significant (Figure 5A), our results indicate that *F. plautii* negatively correlation with the intermediate metabolite Glyceraldehyde ($p < 0.05$, $R = -0.31$, Figure 5D), and Glyceraldehyde's positive correlation with blood glucose levels ($p < 0.001$, $R = 0.85$, Figure 5E) supports the conclusion from Figure 5C. This reinforces the idea that *F. plautii* may affect blood glucose levels through its impact on glyceraldehyde

abundance. However, further larger cohort and animal experiments are needed for validation.” (Line562-572)

(3) Miscellaneous: It is unclear why so few participants' data were included in the study since the cohort being examined is >300 persons. Did the study adhere to the Declaration of Helsinki parameters? Please use "sex" not "gender" since sex is the proper scientific term in this context. Instrument settings, analysis details, and sample preparation details are lacking for the metabolomics methods.

We appreciate the reviewer’s questions and would like to provide further clarification. As mentioned earlier, our initially recruitment included over 300 participants. However, to control for confounding factors that significantly influence the microbiota, such as age and sex, we performed exact matching using the “matchit” function in the R package MatchIt. This rigorous matching process resulted in a reduction of the actual number of subjects in each group. Our cross-sectional study aimed to integrate multi-omics data to explore the mechanisms underlying pre-diabetes. This approach allows us to comprehensively analyze the interplay between different omics layers in the context of pre-diabetes development.

Regarding ethical compliance, this study strictly adhered to the ethical principles outlined in the Declaration of Helsinki. We have provided a detailed summary of the ethical approval process, which includes measures to ensure the protection of participants’ rights and welfare. Specifically, “This study was approved by the Ethics Committee of Guangdong Provincial Hospital of Chinese Medicine (Ethical review number: B2017-199-02). All participants were from Guangdong Provincial Hospital of Chinese Medicine, Fangcun Hospital and gave their written informed consent to publish their clinical data, including all images, clinical data, and other data included in this manuscript.” (Line603-607)

Additionally, in response to the reviewer’s suggestion, we have also replaced “gender” with “sex” throughout the manuscript (Line157, 233, 237, 247) and supplement methodological detail (Line213-221) to ensure scientific accuracy and adherence to appropriate terminology. These revisions enhance the clarity, precision, and ethical standards of our research documentation.

Re: Spectrum01459-24R2 (Integrated Multi-omics Analysis Reveals the Functional Signature of Microbes and Metabolomics in Pre-diabetes individuals)

Dear Dr. Yanmin Liu:

Your manuscript has been accepted, and I am forwarding it to the ASM production staff for publication. Your paper will first be checked to make sure all elements meet the technical requirements. ASM staff will contact you if anything needs to be revised before copyediting and production can begin. Otherwise, you will be notified when your proofs are ready to be viewed.

Sincerely,
Se-Ran Jun
Editor
Microbiology Spectrum